# Detecting molecular interactions in live-cell single-molecule imaging with proximity-assisted photoactivation (PAPA)

**Thomas GW Graham[1], John Joseph Ferrie[1], Gina M Dailey[1], Robert Tjian[1,2]\*, Xavier Darzacq[1]\***

[1]Department of Molecular and Cell Biology, University of California, Berkeley, Berkeley, United States; [2]Howard Hughes Medical Institute, University of California, Berkeley, Berkeley, United States

**Abstract** Single-molecule imaging provides a powerful way to study biochemical processes in live cells, yet it remains challenging to track single molecules while simultaneously detecting their interactions. Here, we describe a novel property of rhodamine dyes, proximity-assisted photoactivation (PAPA), in which one fluorophore (the 'sender') can reactivate a second fluorophore (the 'receiver') from a dark state. PAPA requires proximity between the two fluorophores, yet it operates at a longer average intermolecular distance than Förster resonance energy transfer (FRET). We show that PAPA can be used in live cells both to detect protein–protein interactions and to highlight a subpopulation of labeled protein complexes in which two different labels are in proximity. In proof-of-concept experiments, PAPA detected the expected correlation between androgen receptor self-association and chromatin binding at the single-cell level. These results establish a new way in which a photophysical property of fluorophores can be harnessed to study molecular interactions in single-molecule imaging of live cells.

**\*For correspondence:**
tijcal@berkeley.edu (RT);
darzacq@berkeley.edu (XD)

## Editor's evaluation

This work develops a new method to probe protein–protein interactions using proximity-assisted photo activation, in which a receiver fluorophore (longer wavelength) can be photoactivated by the excitation of a nearby sender fluorophore (shorter wavelength). This new method is validated through in-depth characterization, comparison with FRET, and application to known systems of protein–protein interactions. It will expand the tool kit for probing protein–protein interactions.

## Introduction

Most proteins function by interacting with other proteins, yet we lack tools to study these potentially transient interactions at single-molecule resolution in live cells. Single-particle tracking (SPT) is a valuable approach for monitoring the motions of individual protein molecules (*Chen et al., 2021*; *Hansen et al., 2018*; *Heckert et al., 2022*; *Nguyen et al., 2021*), but it does not distinguish compositionally and functionally distinct complexes of the same protein. Two-color SPT can infer interactions between proteins when both partners are so dilute that they can be fully labeled while still resolving single molecules (*Asher et al., 2021*; *Sotolongo Bellón et al., 2022*; *Wilmes et al., 2020*). For most proteins, however, detection of single molecules requires sparse labeling, which makes double-labeled complexes exceedingly rare. Single-molecule Förster resonance energy transfer (smFRET), though powerful for monitoring intra-molecular conformational changes, is not in general a practical way to detect protein–protein interactions in live cells due to a similar requirement for sparse

**eLife digest** A human body is made up of trillions of cells, each containing millions of proteins working to keep our bodies going. Since the invention of the microscope four hundred years ago, scientists have made large strides in visualizing cells and even single protein molecules within cells. To do this, proteins of interest are labeled with fluorescent dyes that absorb – or are 'excited' by – light of one color, and then give off light of a different color. The labeled proteins are excited by a powerful laser, and a sensitive camera detects the light emitted by single molecules of dye. This technique is called single-particle tracking (SPT), and it can reveal how proteins move around inside a cell.

Because most proteins work together in teams or complexes, it would be useful to track the movement of proteins while at the same time observing their interactions. Unfortunately, SPT does not typically allow scientists to watch how proteins interact with each other. Graham et al. accidentally discovered how to do precisely this.

First, they labeled proteins with two different colored dyes. Then, the dyes were excited using alternating red and green lasers. Repeated excitation destroys the fluorescent dye molecules, and sure enough, red-excited dye molecules went dark over time. Unexpectedly, however, molecules of the dye that had been excited with red light reappeared after exciting the second dye with green light. The fluorescent molecules were not dead, just sleeping. 'Resuscitating' one dye with the other required that they be close together, and therefore this process was called proximity-assisted photoactivation (PAPA for short).

PAPA was able to detect interactions between proteins labeled with different dyes in live human cells, and combining PAPA with SPT allowed Graham et al. to distinguish protein molecules labeled with two different dyes from those labeled with a single dye. Finally, Graham et al. labeled molecules of the androgen receptor protein with two different dyes to monitor how they responded to testosterone. Combining PAPA and SPT measurements successfully detected the pairing of androgen receptor molecules, as well as increased binding of these paired androgen receptor molecules to DNA.

This new way of observing how proteins interact will be useful for studying where and how fast these interactions happen in living cells. Understanding how teams of proteins work together under normal conditions will also shed light on how they misbehave in diseases.

double-labeling, challenges with spectral crosstalk, and the large size of genetically encoded tags relative to the working distance of FRET (see Appendix 1; *Quast and Margeat, 2021*). Fluorescence cross-correlation spectroscopy (FCCS) can detect bulk molecular interactions, yet it does not provide spatial trajectories for individual molecules, which are useful for measuring such properties as chromatin residence time and anomalous diffusion (*Hansen et al., 2020*; *Hansen et al., 2017*; *Izeddin et al., 2014*; *McSwiggen et al., 2019*; *Nguyen et al., 2021*). Bimolecular fluorescence complementation (BiFC) detects molecular interactions based on the reconstitution of a fluorescent protein or HaloTag from two split halves fused to interacting partners (*Ghosh et al., 2000*; *Hu et al., 2002*; *Kerppola, 2008*; *Makhija et al., 2021*; *Shao et al., 2021*). While BiFC can be combined with single-molecule imaging (*Mao et al., 2021*; *Nickerson et al., 2014*; *Shao et al., 2021*), a drawback of this approach is that the extremely strong association of split proteins perturbs the binding equilibrium of their interacting partners (*Kerppola, 2008*; *Kodama and Hu, 2012*; *Nickerson et al., 2014*), making it impossible to accurately measure dynamic interactions.

An alternative in vitro proximity sensor to smFRET was devised by Bates, Blosser, and Zhuang, who observed that exciting one cyanine dye can reactivate a nearby cyanine dye from a dark state (*Bates et al., 2005*). Although photoswitching of cyanine dye pairs enabled early implementations of STORM imaging (*Rust et al., 2006*), its application as a proximity sensor has been limited by the short inter-fluorophore distance required (≤2 nm), the poor cell permeability of cyanine dyes, and the need for high thiol concentrations and an oxygen-scavenging system (*Chen et al., 2016*; *Geertsema et al., 2015*).

To our knowledge, it has not been reported whether a similar process of reactivation can occur for pairs of non-cyanine dyes. However, many fluorophores—notably rhodamine dyes—can enter a dark state and be directly reactivated by short-wavelength (e.g., 405 nm) light (*van de Linde et al., 2011*).

This phenomenon, which has been employed for direct STORM (dSTORM) imaging in both live and fixed cells (*Grimm et al., 2015*; *Heilemann et al., 2008*; *Tang et al., 2021*), is thought to involve conversion of excited triplet-state fluorophores to reduced species whose absorbance is shifted to shorter wavelengths (*Bates et al., 2005*; *Dempsey et al., 2009*; *Gidi et al., 2020*; *Heilemann et al., 2008*; *van de Linde et al., 2011*; *Vaughan et al., 2012*).

The development of bright, cell-permeable Janelia Fluor (JF) dyes, based on rhodamine and silicon-rhodamine chemical scaffolds, has transformed single-molecule imaging in live cells (*Grimm et al., 2015*). Here, we show that Janelia Fluor X 650 (JFX650; *Grimm et al., 2021*) and similar fluorophores can be reactivated from a dark state by excitation of a nearby fluorophore such as Janelia Fluor 549 (JF549), a phenomenon which we term proximity-assisted photoactivation (PAPA). In contrast to cyanine dye reactivation, PAPA of JF dyes occurs under physiological conditions in live cells and requires neither an oxygen-scavenging system nor exogenous thiols. While PAPA requires proximity between the two fluorophores, its effective distance range extends beyond that of FRET, making it a potentially more versatile interaction sensor.

Most importantly, PAPA provides a new way to detect protein interactions in live cells at single-molecule resolution. We show that PAPA can be used to detect the formation of protein dimers and that it can enrich for double-labeled molecules within defined mixtures, albeit not with perfect selectivity (see 'Discussion' and Appendix 2). As a further proof of concept, we combined SPT with PAPA to analyze the increase in chromatin binding induced by self-association of androgen receptor. By enabling the previously elusive detection of protein–protein interactions, PAPA will provide a new dimension of information in live-cell single-molecule imaging.

## Results

### PAPA of JF dyes

We fortuitously discovered PAPA while imaging an oligomeric protein labeled with two different JF dyes. U2OS cells expressing Halo-tagged NPM1 (a pentameric nucleolar protein; *Heckert et al., 2022*) were labeled with a low concentration of Janelia Fluor X 650 HaloTag ligand (JFX650-HTL; *Grimm et al., 2021*) to track single molecules, together with a higher concentration of Janelia Fluor 549 HaloTag ligand (JF549-HTL) to visualize nucleoli. When we alternately excited JFX650 with red light (639 nm) and JF549 with green light (561 nm), we noticed that some JFX650 molecules that had gone dark during red illumination suddenly reappeared after a brief, 7 ms pulse of green light (green vertical lines in *Figure 1ai* and green box in *Figure 1b*, *Figure 1—video 1*). Consistent with previous work (*Grimm et al., 2015*), we also observed reactivation of JFX650 by violet light, both with and without JF549-HTL (violet vertical lines in *Figure 1ai,ii* and violet box in *Figure 1b*). However, reactivation of JFX650 by green light required co-labeling with JF549 (compare *Figure 1ai and ii*), implying that reactivation results not from direct absorption of green light by dark-state JFX650 but indirectly due to excitation of JF549. Green illumination of cells labeled with JF549-HTL alone did not produce localizations in the JFX650 channel, demonstrating that this effect is not due to JF549 photochromism (*Figure 1aiii*).

Because double-labeling of NPM1-Halo pentamers is expected to bring JF549 and JFX650 close together (*Figure 1ai*, right panel), we asked whether proximity of the dyes is required for reactivation. To test this, we expressed fusions of Halo and SNAPf separated by either a short flexible linker (Halo-SNAPf) or a tandem P2A-T2A self-cleaving peptide (Halo-PT2A-SNAPf; *Liu et al., 2017*) in U2OS cells (*Figure 1c*); labeled cells with JF549-HTL and JFX650 SNAP tag ligand (JFX650-STL); and imaged JFX650 with red light interspersed with alternating short pulses of violet and green light. While violet reactivation was similar for both constructs, green reactivation was substantially greater for Halo-SNAPf than for Halo-PT2A-SNAPf, implying that proximity of the two dyes facilitates reactivation by green light (*Figure 1c*). Thus, we term this phenomenon proximity-assisted photoactivation (PAPA). We will call the dye that undergoes reactivation the 'receiver' and the dye whose excitation induces reactivation the 'sender.' Also, we will adopt the terms 'shelving' for conversion of the receiver into the dark state (*Bretschneider et al., 2007*; *Grimm et al., 2015*) and 'direct reactivation' (DR) for reactivation by violet light (*Dempsey et al., 2009*).

Conjugating JFX650 to SNAPf instead of Halo led to more efficient shelving in the dark state, as evidenced by a faster decline in fluorescence during red illumination and greater subsequent

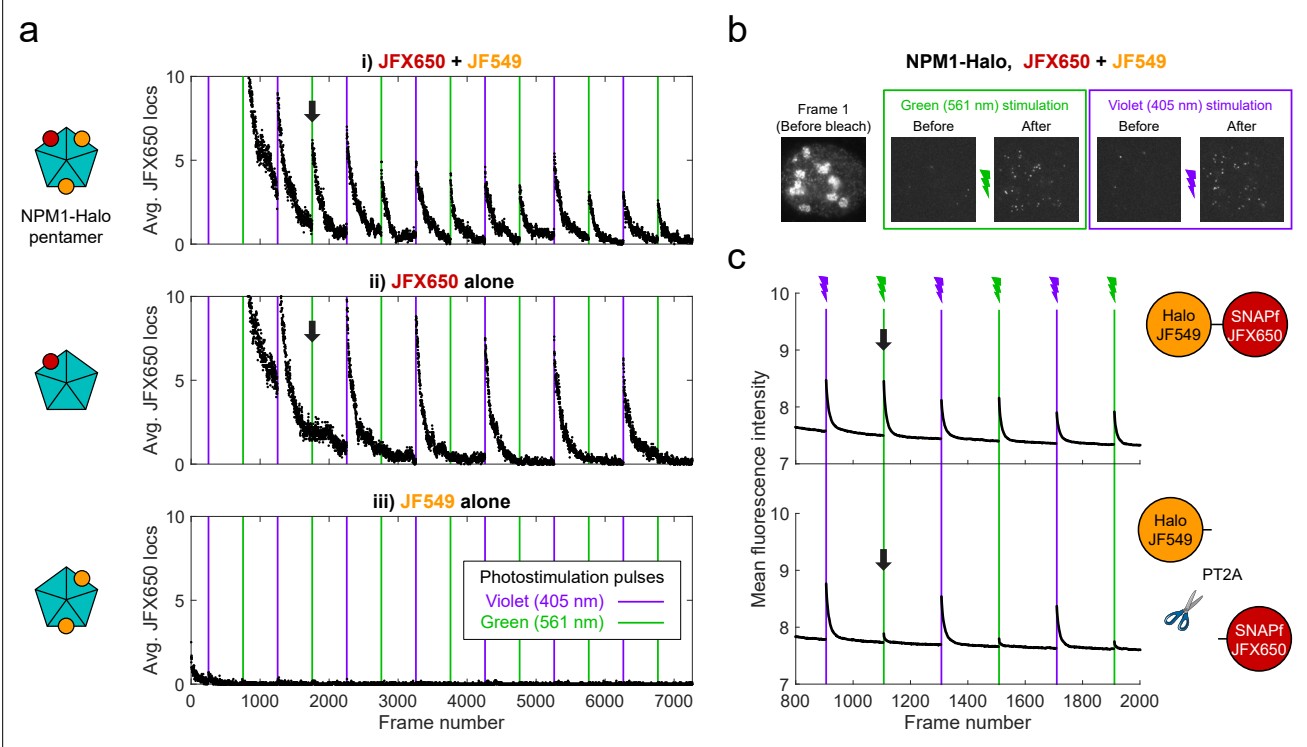

**Figure 1.** Proximity-assisted photoactivation (PAPA) of JFX650 by JF549. (**a**) Green and violet light reactivate JFX650 through distinct JF549-dependent and JF549-independent mechanisms. Left column: schematic of NPM1 pentamers in heterozygously tagged NPM1-Halo U2OS cells labeled with JF549 (orange) and/or JFX650 (red). Right column: average number of localizations in the JFX650 channel as a function of frame number. JFX650 molecules were excited with red (639 nm) light, interspersed with 7 ms pulses of violet (405 nm) and green (561 nm) light (violet and green vertical lines). Reactivation of JFX650 by green light required labeling with JF549 (compare black arrows in i and ii). (**b**) Sample images of a single cell in the JFX650 channel. Leftmost panel: first movie frame prior to fluorophore bleaching/shelving. Green and violet boxes: maximum-intensity projection of all frames immediately before and after green and violet stimulation pulses, showing reactivation of molecules from the dark state. Image dimensions are 24 μm x 24 μm. (**c**) Average fluorescence intensity in the JFX650 channel as a function of frame number in cells expressing a Halo-SNAPf fusion with a flexible linker (top panel; N = 40 cells) or a tandem P2A-T2A self-cleaving peptide between Halo and SNAPf (PT2A; bottom panel; N = 20 cells). Halo was labeled with JF549-HTL and SNAPf with JFX650-STL. Reactivation by violet light pulses (violet lines) occurred in both cases, but reactivation by green light pulses (green lines) was mostly eliminated by the self-cleaving peptide (compare black arrows). Raw intensity traces are displayed without background subtraction.

The online version of this article includes the following video and figure supplement(s) for figure 1:

**Figure supplement 1.** Properties of JFX650 reactivation.

**Figure supplement 2.** Proximity-assisted photoactivation (PAPA) between other sender–receiver pairs.

**Figure supplement 3.** Proximity-assisted photoactivation (PAPA) and direct reactivation (DR) of immobilized single fluorophores.

**Figure 1—video 1.** Initial observation of proximity-assisted photoactivation (PAPA). Heterozygous NPM1-Halo knock-in U2OS cells were labeled sparsely with 50 pM JFX650 HTL and densely with 10 nM JF549 HTL.

https://elifesciences.org/articles/76870/figures#fig1video1

reactivation by violet light (*Figure 1—figure supplement 1a*). This accords with the previous observation that JF549 is more photostable when bound to Halo than when bound to SNAP (*Presman et al., 2017*). Ensemble and single-molecule kinetic measurements indicate that about 10% of JFX650-SNAPf molecules enter the dark state under our experimental conditions and can be reactivated by either DR or PAPA (*Figure 1—figure supplements 1b and 3*). DR by violet light precluded subsequent PAPA by green light, and vice versa, implying that both wavelengths reactivate the same dark state (*Figure 1—figure supplement 1c and d*). We tested other fluorophore pairs and found that PAPA occurred when tetramethylrhodamine (TMR), Janelia Fluor X 549 (JFX549), or Janelia Fluor 526 were used as the sender, or when JF646 or JFX646 were used as the receiver (*Figure 1—figure supplement 2*).

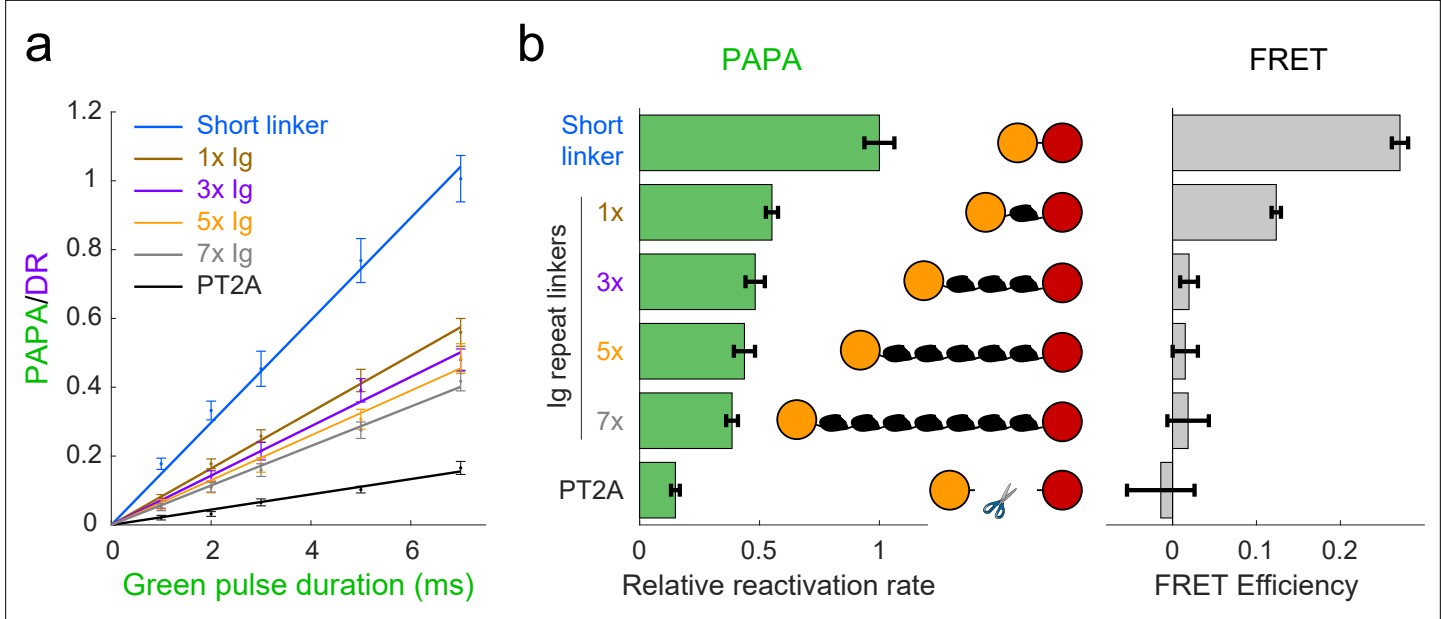

**Figure 2.** Comparison of distance dependence of proximity-assisted photoactivation (PAPA) and Förster resonance energy transfer (FRET). (**a**) PAPA/direct reactivation (DR) ratio vs. green pulse duration for Halo-SNAPf fusions with a short, flexible linker or linkers containing different numbers of tandem Ig domains. Curves are linear fits (y = ax). Error bars, ±2 * SE. PT2A, tandem P2A-T2A self-cleaving peptide. (**b**) Left panel: relative rates of reactivation by PAPA (slope of fits in **a** divided by the slope of the short linker construct). Right panel: FRET efficiency measured using fluorescence lifetime imaging (FLIM).

The online version of this article includes the following figure supplement(s) for figure 2:

**Figure supplement 1.** Simulations and SDS-PAGE analysis of linker constructs.

## Distance dependence of PAPA

To investigate how PAPA depends on sender–receiver distance, we generated fusion transgenes in which Halo and SNAPf were separated by zero, one, three, five, or seven repeats of the titin I91 Ig domain (*Scholl et al., 2016*). The distance distribution between the two dyes was estimated for each fusion protein by simulating an ensemble of conformations using PyRosetta (*Figure 2—figure supplement 1a, b*; *Chaudhury et al., 2010*; *Ferrie and Petersson, 2020*). U2OS cells were stably transfected with each transgene, and fluorescence-activated cell sorting (FACS) was used to obtain pools of cells with similar low expression levels of each protein (*Figure 2—figure supplement 1c and d*; see Appendix 3, Supplementary note 1).

Cells were labeled with a mixture of JF549-HTL and JFX650-STL and imaged as described above with red light interspersed with alternating pulses of violet light to induce DR and green light to induce PAPA. The ratio of the increase in fluorescence intensity in response to green and violet pulses (the 'PAPA/DR ratio') provides a normalized measure of PAPA efficiency, which corrects for cell-to-cell variability in the labeled protein concentration. For sufficiently short reactivation pulses, the PAPA/DR ratio increased linearly with the green pulse duration (with the violet pulse duration held constant), making it possible to measure relative rate constants by linear fitting (*Figure 2a, b*, left panel). In parallel, fluorescence lifetime imaging (FLIM) was used to measure FRET between JF549 and JFX650 for the same fusion proteins (*Figure 2b*, right panel; see Appendix 3, Supplementary note 2).

As predicted by our simulations (*Figure 2—figure supplement 1b*), FRET efficiency between JF549 and JFX650 declined sharply with increasing spacer length, from 0.271 ± 0.010 (95% CI) for the short linker to 0.124 ± 0.006 for a single Ig repeat and 0.020 ± 0.010 for three Ig repeats (*Figure 2b*, right panel). FRET was essentially undetectable for five or seven Ig repeats and for the PT2A self-cleaving peptide linker (*Figure 2b*, right panel). In contrast, PAPA was observed for the 3×, 5×, and 7× Ig linker constructs (*Figure 2a and b*). The rate of photoactivation by green light declined gradually with increasing linker length yet was distinguishable from the background rate of the PT2A self-cleaving linker. These results indicate that PAPA has a less stringent dependence on average inter-fluorophore distance than FRET.

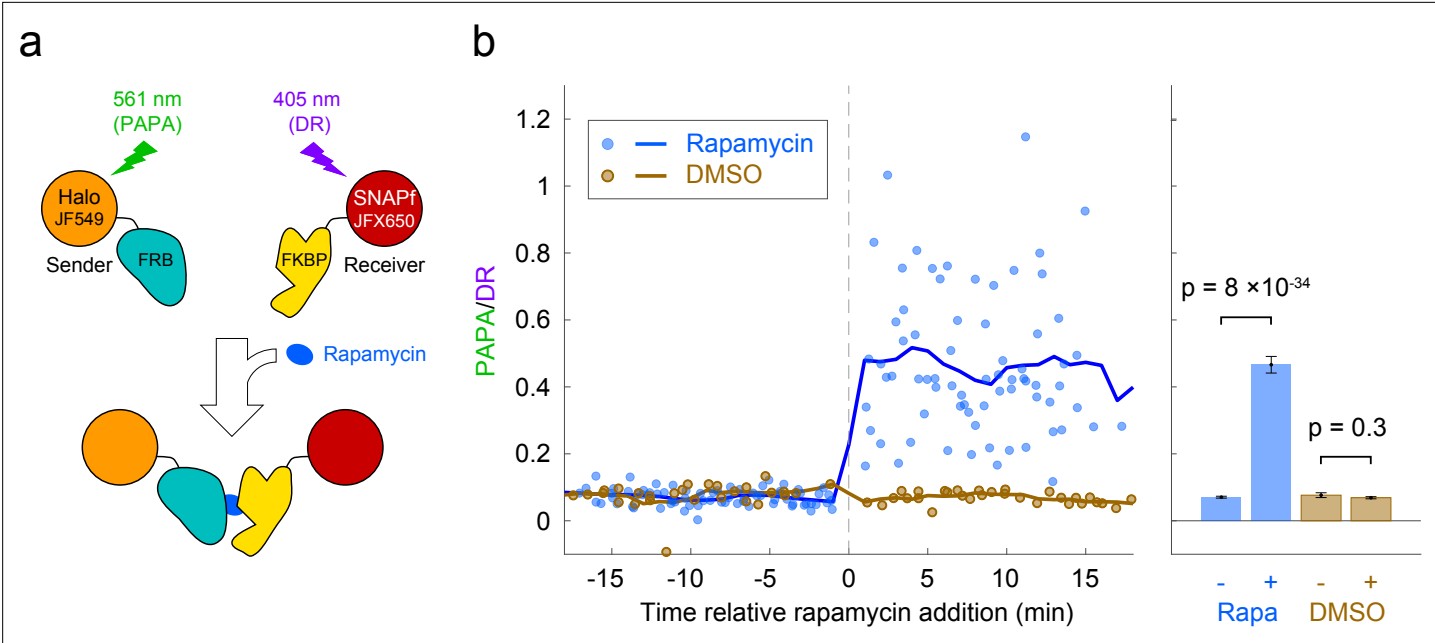

**Figure 3.** Detection of inducible dimerization using proximity-assisted photoactivation (PAPA). (**a**) Halo-FRB was labeled with the sender fluorophore (JF549) and SNAPf-FKBP with the receiver fluorophore (JFX650). After shelving JFX650 with red light, direct reactivation (DR) and PAPA were alternately induced with pulses of violet and green light, respectively. Midway through the experiment, cells were treated with rapamycin (1 μM final concentration) to induce FRB-FKBP dimerization or with dimethylsulfoxide (DMSO) solvent as a negative control. (**b**) Ratio of fluorescence increase due to PAPA (green reactivation) and DR (violet reactivation) as a function of time after rapamycin addition. Blue, rapamycin. Brown, DMSO solvent-only control. Individual data points represent single cells; solid lines show a 2-min moving average. (**c**) Average PAPA/DR ratio before (-) and after (+) addition of rapamycin (Rapa) or DMSO. Total number of cells: 75 before and 74 after rapamycin, 30 before and 30 after DMSO. Error bars, ± 2 * SEM. Statistical significance was calculated using a two-tailed *t*-test.

The online version of this article includes the following video and figure supplement(s) for figure 3:

**Figure supplement 1.** SDS-PAGE and proximity-assisted photoactivation (PAPA) traces of FRB-FKBP.

**Figure 3—video 1.** Direct reactivation and proximity-assisted photoactivation (PAPA) of an inducible protein–protein interaction.

https://elifesciences.org/articles/76870/figures#fig3video1

## Detection of inducible protein–protein interactions using PAPA

Based on the above results, we reasoned that PAPA could be used to detect interaction of two different proteins labeled with SNAPf-JFX650 and Halo-JF549. As a test case, we monitored the rapamycin-inducible interaction of the proteins FRB and FKBP. U2OS cells expressing Halo-FRB and SNAPf-FKBP were labeled with JF549-HTL and JFX650-STL and imaged with alternating green and violet photostimulation as described above (*Figure 3a*, *Figure 3—figure supplement 1a*). Addition of rapamycin caused a dramatic increase in the ratio of PAPA (green reactivation) to DR (violet reactivation), consistent with ligand-induced dimerization of Halo-FRB and SNAPf-FKBP bringing together JF549 and JFX650 (*Figure 3b and c*, *Figure 3—figure supplement 1b*, *Figure 3—video 1*).

## PAPA optically enriches a subset of molecules in defined two-component mixtures

We next asked whether PAPA can be used to spotlight a subpopulation of receiver molecules close to sender molecules. As a simple test case, we analyzed defined mixtures of two proteins—one labeled with JFX650 only, and a second labeled with both JFX650 and JF549—and investigated whether PAPA could optically enrich the double-labeled component to distinguish its properties in single-molecule imaging.

First, we co-expressed SNAPf-tagged histone H2B (SNAPf-H2B), which is predominantly chromatin-bound, along with a Halo-SNAPf fusion with a nuclear localization sequence (Halo-SNAPf-3xNLS), which is mostly unbound (*Figure 4a*; *Hansen et al., 2018*; *Heckert et al., 2022*). Cells were incubated with JFX650-STL and JF549-HTL to label Halo-SNAPf-3xNLS with both JFX650 and JF549 and

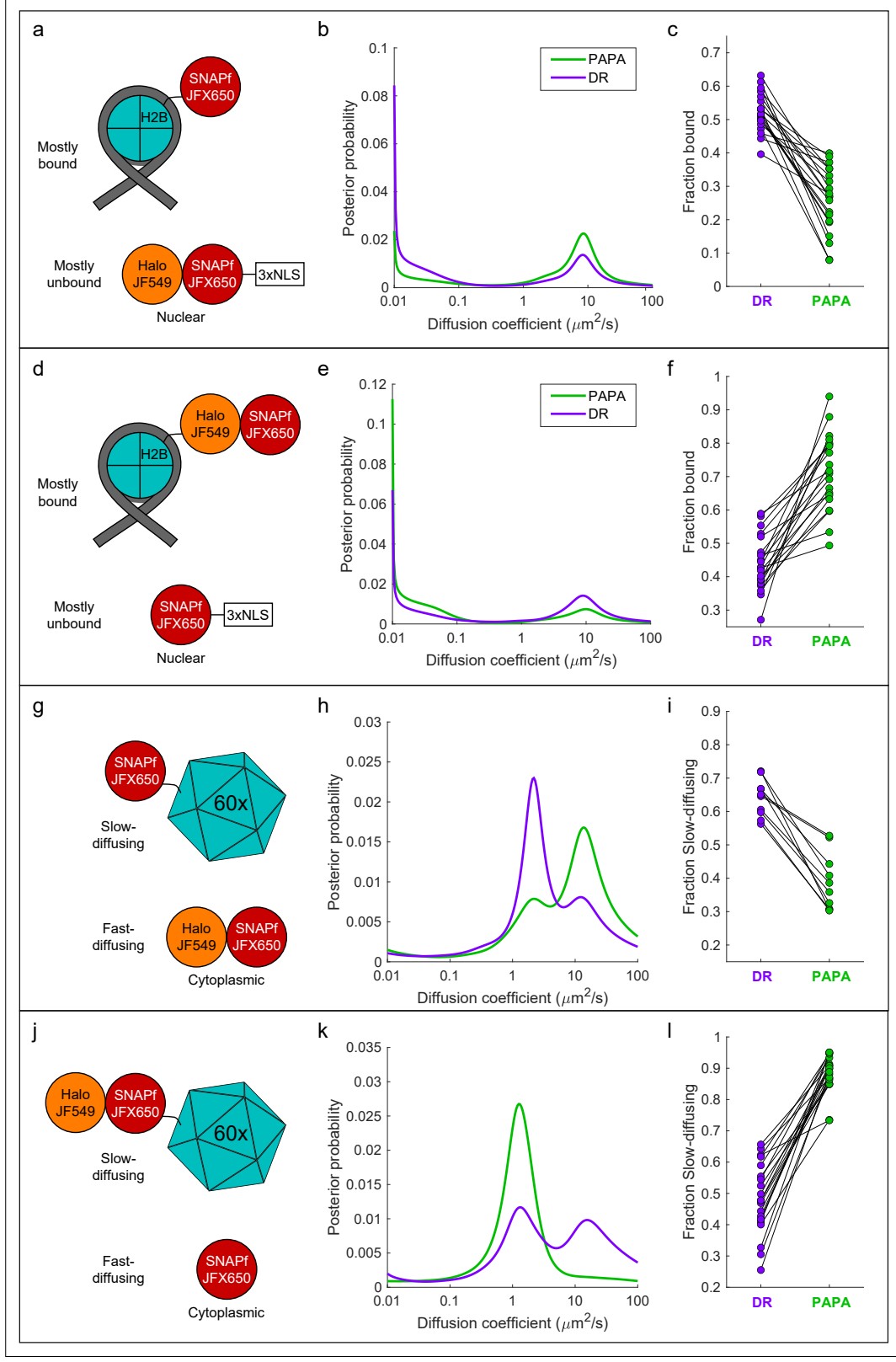

**Figure 4.** 'Unmixing' of defined two-component mixtures using proximity-assisted photoactivation (PAPA). .(**a**) Left column (**a, d, g, j**): schematic of different defined mixtures of two labeled proteins, in which one protein is labeled with JFX650 only and the other is labeled with both JFX650 and JF549. In (**g**) and (**j**), each subunit of the 60-mer is fused to SNAPf or Halo-SNAPf, though only one label is displayed for clarity. (**b**) Center column (**b, e, h,**

*Figure 4 continued on next page*

*Figure 4 continued*

**k**): inferred diffusion spectra of PAPA (green-reactivated) and direct reactivation (DR) (violet-reactivated) trajectories pooled from 20 cells (**b, e, k**) or 10 cells (**h**). (**c**) Right column: fraction bound (**c, f**) or fraction slow-diffusing (**i, l**) of PAPA and DR trajectories from individual cells, obtained from fits to a two-state model (**c, f**) or three-state model (**i, l**). Paired, two-tailed *t*-tests of the comparisons in (**c**), (**f**), (**i**), and (**l**) showed all differences to be statistically significant with p=$9 \times 10^{-9}$, $8 \times 10^{-8}$, $1 \times 10^{-5}$, and $4 \times 10^{-11}$, respectively.

The online version of this article includes the following video and figure supplement(s) for figure 4:

**Figure supplement 1.** SDS-PAGE gels of defined two-component mixtures and one-component controls.

**Figure supplement 2.** Additional analyses of proximity-assisted photoactivation–single-particle tracking (PAPA-SPT) experiment with two-component controls.

**Figure supplement 3.** Proximity-assisted photoactivation–single-particle tracking (PAPA-SPT) analysis of single-component controls.

**Figure supplement 4.** Fitting of two-component diffusion spectra to a mixture of single components.

**Figure 4—video 1.** Proximity-assisted photoactivation–single-particle tracking (PAPA-SPT) experiment of a two-component mixture.

https://elifesciences.org/articles/76870/figures#fig4video1

SNAPf-H2B with JFX650 alone (*Figure 4a*, *Figure 4—figure supplement 1a*). JFX650 fluorophores were thoroughly photobleached/shelved using a 10 s pulse of intense red light, after which JFX650 was imaged with red light interspersed with pulses of green and violet light. After localizing and tracking single molecules, we separated trajectories occurring after a green pulse (PAPA trajectories) from those occurring after a violet pulse (DR trajectories) and applied a recently developed Bayesian state array SPT (saSPT) algorithm (*Heckert et al., 2022*) to infer the underlying distribution of diffusion coefficients for each set of trajectories (its 'diffusion spectrum' for short). As predicted, diffusion spectra revealed two peaks, one corresponding to bound molecules (D = 0.01 µm²/s, the minimum value in the state array), and one corresponding to freely diffusing molecules (D = 8.3 µm²/s). PAPA trajectories were enriched for freely diffusing molecules compared to DR trajectories, as expected if PAPA selectively reactivates JF549/JFX650 double-labeled Halo-SNAPf-3xNLS molecules (*Figure 4b*). Next, PAPA and DR trajectories from individual cells were reanalyzed using a two-state model with bound (D = 0.01 µm²/s) and free (D = 8.3 µm²/s) states. Consistent with the ensemble analysis, PAPA trajectories had a lower bound fraction than DR trajectories in every cell (*Figure 4c*). The same trend is apparent from comparison of particle displacement histograms and raw trajectories (*Figure 4—figure supplement 2a, b*).

To exclude the possibility that enrichment of unbound molecules arose from a systematic bias in our method, we repeated the experiment with the reciprocal mixture of Halo-SNAPf-H2B and SNAPf-3xNLS (*Figure 4d*). As expected, the opposite trend was observed: PAPA trajectories were enriched in bound molecules compared to DR trajectories, both across an ensemble of cells and at the single-cell level (*Figure 4e and f*, *Figure 4—figure supplement 2c and d*). As a further control, we analyzed cells expressing JF549-HTL/JFX650-STL double-labeled Halo-SNAPf-3xNLS or Halo-SNAPf-H2B alone. As expected, PAPA and DR trajectories displayed virtually identical diffusion spectra for these individual components (*Figure 4—figure supplement 3a–f*).

To test whether PAPA can also distinguish a mixture of diffusing components, we co-expressed fast-diffusing cytosolic Halo-SNAPf with a SNAPf-tagged synthetic protein that forms large, slowly diffusing 60-mers (*Hsia et al., 2016*; *Figure 4g*). As expected, diffusion spectra had two peaks corresponding to slow-diffusing (SNAPf-60-mer) and fast-diffusing (Halo-SNAPf) components (*Figure 4h*). Compared to DR trajectories (violet curve), PAPA trajectories (green curve) were strongly enriched in the fast-diffusing subpopulation, consistent with selective reactivation of the double-labeled Halo-SNAPf protein by green light (*Figure 4h*). The same trend was observed in single-cell reanalysis, displacement histograms, and raw trajectories (*Figure 4i*, *Figure 4—figure supplement 2e and f*). The enrichment of the fast-diffusing population was not absolute as a slow-diffusing shoulder peak was still observed among PAPA trajectories (*Figure 4h*, green curve; see 'Discussion'). As before, swapping SNAPf and Halo-SNAPf labels yielded the opposite trend, both at the ensemble and single-cell level (*Figure 4j–l*, *Figure 4—figure supplement 2g and h*, *Figure 4—video 1*). Enrichment of the 60-mer peak by PAPA is especially pronounced in this case (compare green and violet curves in

*Figure 4k*), which may reflect reactivation of a receiver molecule by any of several neighboring sender molecules within a 60-mer.

To estimate the fold enrichment of double-labeled molecules by PAPA, the curves shown in *Figure 4* were fitted to a linear combination of the distributions for individual components. For Halo-SNAPf-3xNLS + SNAPf-H2B, the best fit was obtained for DR trajectories with a mixture of 53% Halo-SNAPf-3xNLS and 47% SNAPf-H2B, while the best fit for PAPA trajectories was obtained with 91% Halo-SNAPf-3xNLS and 9% SNAPf-H2B (*Figure 4—figure supplement 4a*). The estimated ratio of Halo-SNAPf-3xNLS to SNAPf-H2B thus increases from 0.53/0.47 ≈ 1.1 among DR trajectories to 0.91/0.09 ≈ 10 among PAPA trajectories, an approximately ninefold enrichment of double-labeled molecules by PAPA. Similar calculations showed that PAPA enriched Halo-SNAPf-H2B over SNAPf-3xNLS by 3.7-fold (*Figure 4—figure supplement 4b*), Halo-SNAPf over SNAPf-60mer by 7.6-fold (*Figure 4—figure supplement 4c*), and Halo-SNAPf-60mer over SNAPf by 37-fold (*Figure 4—figure supplement 4d*).

Taken together, these results demonstrate that PAPA can be used to enrich a subpopulation of molecules in which a receiver fluorophore (e.g., JFX650) is in proximity to a sender fluorophore (e.g., JF549), thereby revealing the distinct properties of this subpopulation at both the ensemble and single-cell level. While this enrichment was substantial—between 3.7- and 37-fold for different defined mixtures—it was not absolute, and thus it is crucial not to misinterpret PAPA trajectories as a pure sample of interacting molecules (see 'Discussion' and Appendix 2).

## Distinguishing the properties of androgen receptor monomers and dimers in single cells

As a proof-of-concept biological application, we tested whether PAPA could be used to detect ligand-induced self-association of mammalian androgen receptor (AR) and distinguish the properties of AR monomers and dimers/oligomers. First, we stably co-expressed SNAPf and Halo fusions of mouse AR in U2OS cells (which express very little endogenous AR [*Dellal et al., 2020*]), labeled the two proteins with a mixture of JFX650-STL and JF549-HTL (*Figure 5—figure supplement 1a*), and measured the PAPA/DR ratio by quantifying changes in JFX650 fluorescence intensity in response to alternating green and violet stimulation as described above. As expected, treatment with the androgen dihydro-testosterone (DHT) led to an increase in the ratio of PAPA to DR over the course of several minutes (*Figure 5b*, *Figure 5—figure supplement 1b*), consistent with the two fluorophores being brought together by ligand-induced interaction between SNAPf-mAR and Halo-mAR.

Next, we combined PAPA with single-molecule imaging to assess how self-association influences diffusion and chromatin binding by AR. Consistent with previous biochemical and live-cell imaging experiments, addition of DHT caused an increase in the overall bound fraction of AR (*Figure 5c and d*; *Schaufele et al., 2005*; *van Royen et al., 2007*; *van Royen et al., 2012*). Strikingly, PAPA trajectories had a higher bound fraction than DR trajectories, both before and after addition of DHT (*Figure 5c, d*). This is consistent with an increase in the affinity of AR for specific DNA sequence motifs upon self-association. Moreover, PAPA revealed that a subset of AR molecules self-associated and bound chromatin with elevated affinity even prior to addition of exogenous androgen. Thus, PAPA can be applied to monitor regulation of a biologically important protein–protein interaction in live cells, discern its effect on chromatin binding, and reveal the existence of molecular subpopulations.

## Discussion

Here, we have described a novel and useful property of rhodamine dyes, PAPA, in which excitation of a 'sender' fluorophore (e.g., JF549) reactivates a nearby 'receiver' fluorophore (e.g., JFX650) from a dark state. By enabling targeted reactivation of receiver fluorophores near a sender fluorophore, PAPA provides a new way to detect molecular interactions in live cells (*Figure 6*): First, Halo-tagged proteins are labeled with the sender fluorophore and SNAPf-tagged proteins with the receiver fluorophore. Second, cells are illuminated with intense red light to shelve receiver fluorophores in the dark state. Third, alternating pulses of green and violet light are applied to reactivate receiver fluorophores by PAPA and DR, respectively, and these reactivated fluorophores are imaged using red illumination. The ratio of green to violet reactivation provides a measure of protein–protein interaction, while analysis of green-reactivated and violet-reactivated single-particle trajectories makes it possible to

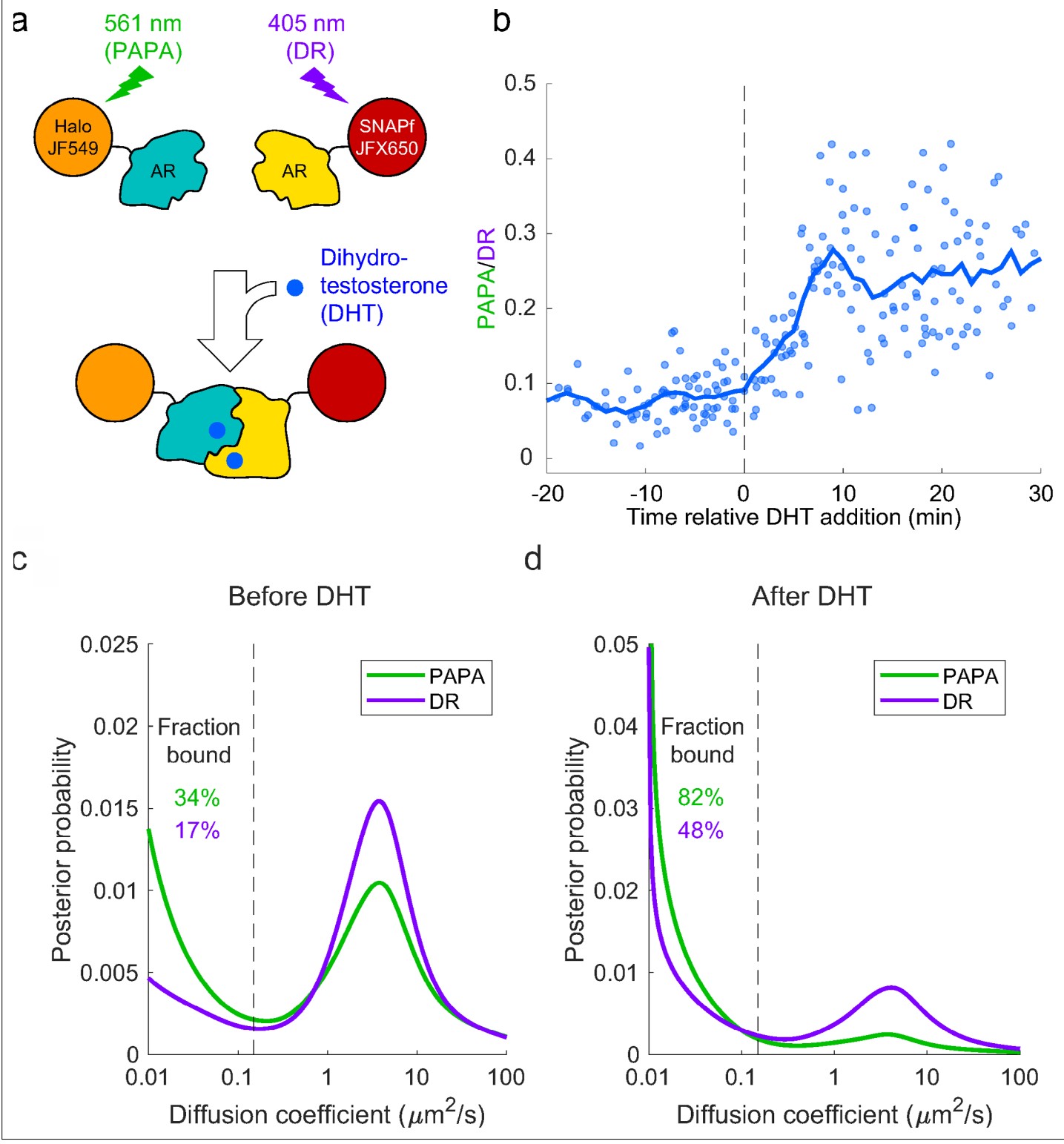

**Figure 5.** Analysis of mammalian androgen receptor using proximity-assisted photoactivation–single-particle tracking (PAPA-SPT). (**a**) Schematic of dihydrotestosterone (DHT)-induced dimerization of JF549-Halo-mAR and JFX650-SNAPf-mAR. (**b**) PAPA/direct reactivation (DR) ratio as a function of time relative DHT addition. (**c**) Diffusion spectra of PAPA and DR trajectories. (**c**) Before addition of DHT; N = 55 cells. (**d**) After addition of DHT to a final concentration of 10 nM; N = 81 cells. Fraction bound was quantified by summing the portion of each curve below D = 0.15 µm²/s (vertical dashed line).

The online version of this article includes the following figure supplement(s) for figure 5:

**Figure supplement 1.** Proximity-assisted photoactivation (PAPA) analysis of androgen receptor.

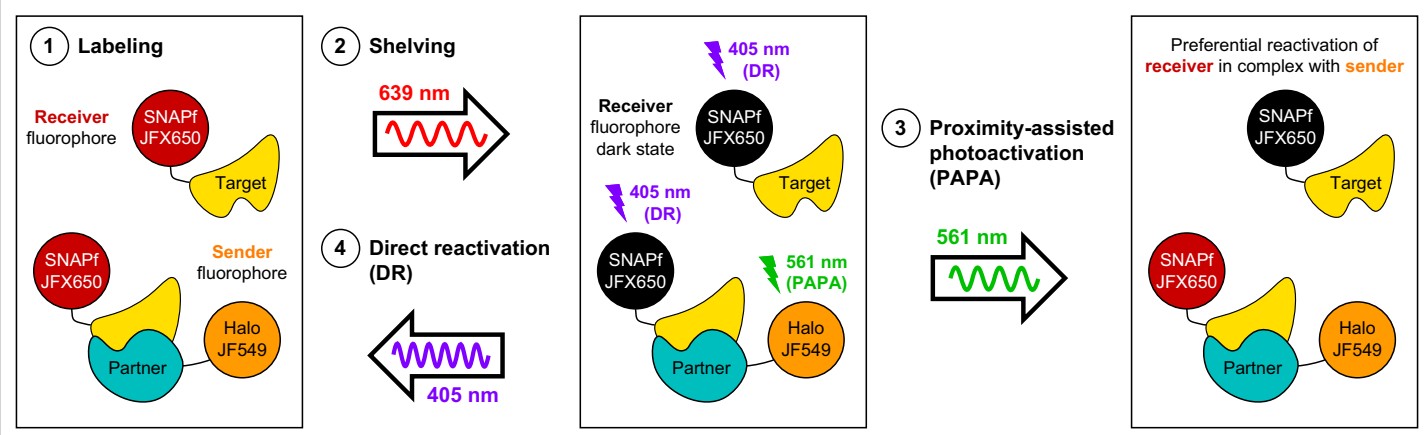

**Figure 6.** Using proximity-assisted photoactivation (PAPA) to spotlight protein–protein interactions. (1) Label a SNAPf-tagged Target protein with a receiver fluorophore (e.g., JFX650) and a Halo-tagged Partner protein with a sender fluorophore (e.g., JF549). (2) Shelve the receiver fluorophore in the dark state using intense 639 nm illumination. Image receiver molecules with 639 nm light while alternately illuminating with (3) pulses of 561 nm light to induce PAPA of receiver-labeled Target molecules in complex with sender-labeled Partner molecules, and (4) pulses of 405 nm light to induce direct reactivation (DR) of receiver fluorophores, independent of proximity to the sender.

compare the overall population of receiver-labeled molecules (violet; DR) to a subpopulation enriched for double-labeled complexes (green; PAPA).

While the physical mechanism underlying PAPA remains unclear, its more flexible distance dependence than either FRET (*Figure 2*) or cyanine dye photoswitching (*Bates et al., 2005*) suggests a distinct process. One speculative hypothesis is that the excited sender reacts with some other molecule in the cell, producing a short-lived chemical species that diffuses a limited distance to react with and reactivate the receiver dark state. However, we cannot exclude an alternative model proposed by Gidi and colleagues for cyanine dye pairs, in which the absorbance spectrum of the receiver dark state has a broad tail toward longer wavelengths, allowing the sender to serve as an 'antenna' that facilitates receiver reactivation via some form of energy transfer (*Gidi et al., 2020*). Even if such a process had a low quantum yield, the all-or-nothing property of reactivation might make it observable experimentally. PAPA can complement other techniques for monitoring molecular interactions. Although smFRET is useful for measuring distances between fluorophores in vitro, PAPA provides multiple advantages for live-cell imaging: first, PAPA circumvents the trade-off between labeling density and spectral crosstalk inherent in smFRET (see Appendix 1). It is impractical to detect molecular interactions in cells by sparsely labeling both the FRET donor and acceptor, as double-labeled complexes will be vanishingly rare. Attempting to solve this problem by densely labeling either the donor or the acceptor creates the new problem of fluorescence bleed-through from the densely labeled channel, which may be orders of magnitude brighter than signal from the sparsely labeled channel. In PAPA, sender and receiver excitation occur at different times, eliminating fluorescence bleed-through from the sender into the receiver channel. Hence, one interacting partner can be sparsely labeled with the receiver and the other densely labeled with the sender, permitting efficient detection of double-labeled complexes. Second, PAPA has a conveniently longer working distance than FRET (*Figure 2*), which might be extended further by elongating the linkers between Halo/SNAPf and the protein of interest. Because photoactivation is an all-or-nothing event, a signal can in principle be detected if even a fraction of linker conformations orients the dyes close enough together for PAPA to occur. Unlike single-molecule BiFC, PAPA does not perturb binding equilibria, making it possible to use PAPA to study transient, reversible interactions.

Although PAPA significantly enriches for complexes double-labeled with sender and receiver, its selectivity is not perfect (see Appendix 2). A background level of PAPA was still observed when Halo and SNAPf were separated by a self-cleaving peptide tag (*Figures 1c and 2a and b*), and this cannot be explained either by incomplete cleavage (*Figure 2—figure supplement 1c and d*) or by direct reactivation of JFX650 by 561 nm light (*Figure 1a, Figure 1—figure supplement 2d*). Moreover, some 'contamination' of PAPA trajectories with single-labeled molecules was evident in experiments with defined two-component mixtures (e.g., slower-diffusing peak in green curve of *Figure 4h*). When

interpreting PAPA-SPT experiments, it is thus critical to keep in mind that green-reactivated trajectories, although enriched for sender–receiver complexes, will inevitably be contaminated by some level of nonspecific background (see Appendix 2). PAPA provides enrichment—not purification—of double-labeled complexes.

Nonspecific background in PAPA-SPT experiments could arise from multiple sources (see Appendix 2): first, even molecules that are not physically associated come into proximity by chance at some rate. Indeed, when JFX650-labeled cells were bathed in high concentrations of free JF549 dye, reactivation by green light occurred in proportion to the JF549 concentration (*Appendix 2—figure 1*). Nonspecific background is thus expected to be greater when sender-labeled proteins are expressed at high levels. Second, dark-state fluorophores spontaneously reactivate at a low basal rate even without photostimulation (*Grimm et al., 2015*; *Tang et al., 2021*). Third, although we chose a time interval between green and violet pulses sufficient to bleach or re-shelve most reactivated fluorophores, it is possible that a small fraction of fluorophores reactivated by a violet pulse survived until the subsequent green pulse. Modeling of these different background contributions will be required to quantify more precisely the characteristics of interacting and noninteracting molecular subpopulations.

Notwithstanding these technical imperfections, PAPA has the potential to open new experimental routes toward understanding the dynamics of protein complexes in live cells. Our results show that PAPA can be used to detect potentially transient protein–protein interactions (*Figure 3*, *Figure 3—figure supplement 1*) and to infer the composition of different peaks in single-molecule diffusion spectra (*Figure 4*, *Figure 4—figure supplement 2*). A proof-of-concept application to mammalian AR revealed at a single-cell level the relationship between AR self-association and chromatin binding (*Figure 5*, *Figure 5—figure supplement 1*). Future applications of PAPA could include measuring differences in the chromatin residence time of different transcriptional subcomplexes, detecting transient interactions mediated by low-complexity domains, or assessing the consequences of posttranslational modifications such as SUMOylation. In principle, a 'pulse-chase' PAPA experiment could be used to measure dissociation and binding kinetics of molecular complexes in live cells by monitoring how the diffusion spectrum of reactivated molecules changes as a function of time after the reactivation pulse. Although this study involved labeled proteins, PAPA could potentially be used to detect interactions between other biomolecules as well. By revealing the distinct features of specific molecular complexes, PAPA will provide a powerful new tool to probe biochemical mechanisms in live cells.

## Methods

### Cell culture

U2OS cells were grown in Dulbecco's modified Eagle's medium (DMEM) with 4.5 g/l glucose (Thermo Fisher #10566016), 10% fetal bovine serum (FBS), and 100 U/ml penicillin-streptomycin (Thermo Fisher #15140122) at 37°C and 5% $CO_2$. Phenol red-containing medium was used for propagation of cells, while phenol red-free medium (Thermo Fisher #21063029) was used to minimize fluorescence background in imaging experiments.

### Cloning

Ig linkers were subcloned from a previously described plasmid containing repeats of the titin I91 Ig domain, which were codon-shuffled to prevent recombination (*Scholl et al., 2016*). The various Halo and SNAPf fusion constructs described in this article were generated by PCR and isothermal assembly, and all constructs were completely sequenced before use. Two-component expression plasmids included a codon-shuffled SNAPf-3xNLS-T2A-P2A cassette that was ordered as a gBlock from Integrated DNA Technologies (IDT). All plasmid sequences are available at https://gitlab.com/tgwgraham/papa_paper_plasmids, copy archived at swh:1:rev:984803582c0023a9fe975ec6d41be3d723c2737b; *Graham, 2022a*.

### Stable transformation of cells and selection of clonal lines

To generate stable lines by PiggyBac integration, U2OS cells from a confluent 10 cm plate were trypsinized, resuspended in DMEM, and divided between two 15 ml conical tubes. Cells were centrifuged for 2 min at 200 × *g*, and the medium was aspirated and replaced with 100 µl of Lonza Kit V transfection reagent (82 µl of Kit V solution and 18 µl of Supplement I; Cat# VCA-1003) containing 1 µg of the

donor plasmid and 1 µg of Super PiggyBac transposase plasmid. The cell suspension was transferred to an electroporation cuvette and electroporated using program X-001 on an Amaxa Nucleofector II (Lonza). Cells in the cuvette were mixed with 300 µl of DMEM, and 100 µl of the cell suspension was diluted in 10 ml of DMEM in a 10 cm plate. After allowing cells to grow for 1–2 days, selection was initiated by adding puromycin to a final concentration of 1 µg/ml.

To generate clonal cell lines of Halo-mAR + SNAPf-mAR and FKBP-SNAPf-3xNLS + FRB-Halo-3xNLS, a polyclonal pool of stably transfected cells from a 10 cm plate was labeled with a mixture of 50 nM JF549 SNAP tag ligand and 50 nM JFX650 HaloTag ligand, and FACS was used to sort single cells expressing both proteins into separate wells of a 96-well plate. For pTG800 (3xFlag-Halo-SNAPf-3xNLS-T2A-P2A-H2B-SNAPf-3xNLS), single-cell clones were obtained by limiting dilution into 96-well plates. Polyclonal pools of stably transfected cells were used for the other two-component and one-component experiments in *Figure 4* and *Figure 4—figure supplements 1–4*.

For the experiments in *Figure 2* and *Figure 2—figure supplement 1*, FACS was used to obtain polyclonal pools of U2OS cells expressing a low level of each Halo-linker-SNAPf construct. Confluent 10 cm plates of cells were stained with 50 nM each of JF549-STL and JFX650-HTL, and cells were sorted using the same intensity gate in the JFX650-Halo channel. Cells expressing pTG820 (Halo-3x Ig-SNAPf) and pTG828 (Halo-5x Ig-SNAPf) were sorted on a different day using the intensity of the previously sorted pTG747/U2OS pool to define a gate in the JFX650-Halo channel.

## Visualization of fluorescently labeled proteins by SDS-PAGE

Cells in either 10 cm plates or 6-well plates were labeled with 500 nM of the indicated HTL or STL ligand for 1 hr at 37°C, washed twice with 1× PBS, trypsinized, and resuspended in DMEM. Cells were counted using a Countess 3 FL cell counter (Invitrogen), pelleted by centrifugation for 2 min at 200 × g, and frozen at –80°C. Lysates were prepared by addition of 1 ml (for 10 cm plates) or 200 µl (for 6-well plates) of SDS lysis buffer without dye (*Cattoglio et al., 2019*). Each lysate was passed through a 26-gauge needle 10 times to reduce its viscosity.

Custom 8-well, 1.5 mm combs for SDS-PAGE were 3D-printed using an AnyCubic Photon 3D printer (model files available at https://gitlab.com/tgwgraham/gel-combs; copy archived at swh:1:rev:-7d5a083eec963534c1bf54632e5e3fb7606e5518; *Graham, 2022b*). For the gels in *Figure 2—figure supplement 1c and d*, samples of cell lysate corresponding to 100,000 cells were separated on a 10% SDS-PAGE gel, which was imaged on a Pharos FX imager (Bio-Rad) using the 'low-intensity' setting in the Cy3 channel. Cell lysate corresponding to 60,000 cells was loaded per lane of the gels in *Figure 3—figure supplement 1*, *Figure 4—figure supplement 1*, and *Figure 5—figure supplement 1*. The gel in *Figure 4—figure supplement 1a* was imaged on a Pharos FX imager (Bio-Rad) using the 'low-intensity' setting in the Cy5 channel. The gels in *Figure 3—figure supplement 1a*, *Figure 4—figure supplement 1b*, and *Figure 5—figure supplement 1a* were imaged in the 700 nm channel on an Odyssey imager (LI-COR) at 169 µm resolution with the 'medium' quality setting and a z-offset of 0.5 mm. Precision Plus Protein All Blue Prestained Protein Standards (Bio-Rad #1610373) were used as molecular weight standards for all gels.

## Live-cell single-molecule imaging

One day prior to imaging, 25 mm No. 1.5H glass coverslips (Marienfeld, #0117650) were immersed in isopropanol, transferred with forceps to 6-well plates, and aspirated thoroughly to remove all traces of isopropanol. Cells were trypsinized, counted using a Countess 3 FL cell counter (Invitrogen), centrifuged for 2 min at 200 × g, resuspended in phenol red-free DMEM, and plated at a density of 5 × $10^5$ cells per well. Just prior to imaging, cells were incubated with Janelia Fluor HaloTag and SNAP tag ligands in phenol red-free DMEM for 15 min at 37°C, washed twice with 1× phosphate-buffered saline, and destained for at least 15 min in phenol red-free DMEM. The following dye concentrations were used for staining:

- 10 nM JF549-HTL and/or 250 pM JFX650-HTL for *Figure 1a and b*.
- 50 nM JF549-HTL and 5 nM JFX650-STL for *Figure 1c*, *Figure 2a and b*, *Figure 3*, *Figure 4*, *Figure 5*, and *Figure 1—figure supplement 1b–d*, *Figure 3—figure supplement 1b*, *Figure 4—figure supplements 1–4*, *Figure 5—figure supplement 1*, and *Appendix 2—figure 1* (dashed blue line).
- 50 pM JFX650 HTL or 5 nM JFX650-STL for *Figure 1—figure supplement 1a*.

- 50 nM JF526/JF549/JFX549/TMR-HTL and/or 5 nM JF646/JFX650-STL for *Figure 1—figure supplement 2*.

Coverslips were mounted in a stainless steel Attofluor Cell Chamber (Thermo Fisher #A7816) and covered with 1 ml of phenol red-free DMEM with 10% FBS and penicillin/streptomycin. Cells were imaged using HILO illumination on the microscope described in detail in *Hansen et al., 2018*. Laser power densities used for imaging were approximately 52 W/cm$^2$ for 405 nm (violet), 100 W/cm$^2$ for 561 nm (green), and 2.3 kW/cm$^2$ for 639 nm (red). Fluorescence emission was filtered through a Semrock 676/37 bandpass filter.

For the experiments in *Appendix 1—figure 1*, a Di02-R635 dichroic (Semrock) was used to separate JF549 and JFX650 emission, which were filtered through 593/40 and 676/37 bandpass filters, respectively, and imaged on separate cameras.

Cells were imaged at a rate of 7.48 ms/frame. Different experiments employed variations of an illumination sequence with alternating pulses of 639 nm red (R), 561 nm green (G), and 405 nm violet (V) light synchronized to the camera. We use these abbreviations below and indicate the duration of the light pulse in brackets. For instance, '250 R [2 ms]' denotes 250 frames with a 2 ms pulse of red 639 nm illumination per frame. Red illumination was restricted to one 2 ms stroboscopic pulse per frame in single-molecule tracking to reduce the motion blur of moving molecules (*Hansen et al., 2018*). The green and violet pulse durations were adjusted in different experiments to maintain a trackable density of localizations after each pulse.

- *Figures 1a and b, 3 and 5b*, *Figure 3—figure supplement 1b*, *Figure 5—figure supplement 1b*: Ten cycles of 250 R [2 ms], 1 V [7 ms], 500 R [2 ms], 1 G [7 ms], 250 R [2 ms]. The number of cycles in *Figures 3 and 5b* was reduced to 4 and 5, respectively.
- *Figure 1c* and *Figure 2*: Five cycles of 100 R [7 ms], 1 V [7 ms] + R [7 ms], 200 R [7 ms], 1 G [7 ms] + R [7 ms], 100 R [7 ms].
- *Figures 4–6*, *Figure 4a–f*, *Figure 5c and d*, *Figure 5—figure supplement 1*, and *Figure 3—figure supplement 1*, *Figure 4—figure supplements 2 and 3a-f*: Cells were first illuminated 10 s with 639 nm light to either photobleach or shelve most JFX650 fluorophores and then imaged with 10 cycles of 250 R [2 ms], 1 V [0.5 ms]+R [2 ms], 500 R [2 ms], 1 G [X ms] + R [2 ms], 250 R [2 ms]. Owing to differences in the PAPA efficiency and protein concentration between samples, the green pulse duration (X) was adjusted empirically to obtain a trackable number of localizations following photostimulation. It was set to 0.5 ms for *Figure 4g–l* and *Figure 4—figure supplement 2e–h* and *Figure 4—figure supplement 3g–l*; 2 ms for *Figures 4a–f and 5d*, *Figure 5—figure supplement 1c* (after DHT), and *Figure 4—figure supplement 2a–d* and *Figure 4—figure supplement 3a–f*; and 7 ms for *Figure 5c* and *Figure 5—figure supplement 1c*. Because DHT addition greatly increased the PAPA efficiency in AR experiments (*Figure 5b*), the green pulse duration was shortened from 7 ms before DHT to 2 ms after DHT in *Figure 5c and d* and *Figure 5—figure supplement 1c* to keep the number of localizations per frame roughly equivalent and prevent PAPA trajectories from becoming too dense for accurate tracking.
- *Figure 1—figure supplement 1a*: Ten cycles of 100 R [7 ms], 1 V [7 ms] + R [7 ms], 100 R [7 ms]. Only the first two cycles are shown in the figure.

## Analysis of ensemble PAPA experiments

For ensemble PAPA experiments (*Figure 1c*, *Figure 2a and b*, *Figure 3b and c*, *Figure 5b*, *Figure 1—figure supplements 1–2*, *Figure 3—figure supplement 1b*, *Figure 5—figure supplement 1b*, and *Appendix 2—figure 1*), custom MATLAB code was used to sum the total intensity of all pixels in the field of view at each frame. Frame-by-frame intensity across multiple movies was averaged to obtain 'sawtooth' plots of intensity vs. frame number (*Figure 1c* and *Figure 1—figure supplement 1a,c,and d*, *Figure 1—figure supplement 2*, *Figure 3—figure supplement 1b*, and *Figure 5—figure supplement 1b*). The PAPA/DR ratio was calculated for *Figure 2a and b*, *Figure 3b and c*, *Figure 5b*, and *Appendix 2—figure 1* by dividing the mean increase in fluorescence intensity induced by green and violet pulses. For *Figure 2a and b*, which used an illumination sequence with fewer frames per cycle, the initial green and violet pulse were omitted from the averages to avoid the transient photobleaching/shelving phase at the beginning of each movie.

## Bulk measurement of reactivation kinetics

To measure direct reactivation of JFX650-SNAPf as a function of 405 nm illumination time (*Figure 1—figure supplement 1b*), cells expressing Halo-SNAPf-3xNLS (pTG747) were stained with 5 nM JFX650 STL and 50 nM JF549 HTL and imaged at 7.48 ms/frame using a five-phase protocol: (1) 20 frames with 1 ms pulses of 639 nm (intensity measurement before bleaching/shelving), (2) 400 frames with 7 ms pulses of 639 nm (bleaching/shelving), (3) 20 frames with 1 ms pulses of 639 nm (intensity measurement after bleaching/shelving), (4) N frames with 7 ms pulses of 405 nm (reactivation), and (5) 20 frames with 1 ms pulses of 639 nm (intensity measurement after reactivation). The total pixel intensity was summed for all 20 frames in phases (1), (3), and (5), and the fractional reactivation was calculated by subtracting the increase in signal between (3) and (5) by the initial drop in signal between (1) and (3). The number of violet frames in phase 4, N, was varied as indicated by the values on the horizontal axis of *Figure 1—figure supplement 1b*. The data were fitted to a single-exponential model.

## Single-molecule measurement of reactivation

To monitor reactivation of single immobilized JFX650 fluorophores, U2OS cells expressing H2B-Halo-SNAPf were labeled for 15 min with 50 pM of JFX650 STL, either with or without 50 nM JF549 HTL. Cells were imaged at 7.48 ms/frame with five cycles of 100 frames of 639 nm illumination, 10 frames of either 561 nm or 405 nm illumination, and another 100 frames of 639 nm illumination. Individual, well-separated JFX650 fluorophores (≥8 pixels apart) were identified in the first frame of each movie, and each fluorophore was scored as fluorescent in subsequent frames if a localization was detected within 4 pixels of its initial position.

## FLIM-FRET

Fluorescence lifetime was measured on a Zeiss LSM 980 confocal microscope equipped with a Becker & Hickl SPC-150NX TCSPC module. Cells were labeled with a mixture of 50 nM JFX650-HTL and 50 nM JF549-STL for 1 hr at 37°C, or with 50 nM JF549-STL alone as a no-FRET control. After briefly washing twice with 1× PBS, cells were destained for at least 15 min prior to imaging. JF549 was excited using a 562 nm laser, and fluorescence emission was filtered through a Semrock 593/40 band-pass filter. Signal was acquired for 10 s over a 256 × 256 px region centered on each cell nucleus. Raw data were imported into SPCImage (Becker & Hickl) and fitted to a single-exponential model without binning pixels. Decay constants and total fluorescence intensities for each pixel were exported in .csv format. Custom MATLAB code was used to define nuclear masks by intensity thresholding and determine the mean fluorescence lifetime within the nucleus. FRET efficiency was calculated using the formula $E_{FRET} = 1 - \tau / \tau_0$, where $\tau$ is the fluorescence lifetime of the sample and $\tau_0$ is the fluorescence lifetime of cells stained with JF549-STL donor only.

## Measuring mutual 'occlusion' of PAPA and DR

To measure whether PAPA precludes DR and vice versa (*Figure 1—figure supplement 1c and d*), we first labeled cells expressing Halo-SNAPf-3xNLS (pTG747) with 5 nM JFX650 STL and 50 nM JF549. We then imaged at 7.48 ms/frame in three phases: (1) 500 frames of 639 nm light (7 ms pulses), alternating with unrecorded frames with either no illumination (black curves) or 7 ms pulses of 405 nm (violet curves) or 561 nm (green curves) illumination; (2) 20 frames with 7 ms pulses of 405 nm light (*Figure 1—figure supplement 1c*) or 561 nm light (*Figure 1—figure supplement 1d*); and (3) 100 frames of 639 nm light (7 ms pulses). Fluorescence intensity traces were prepared as described in 'Analysis of ensemble PAPA experiments' above.

## Analysis of SPT data

Particles were localized and tracked using the quot package (https://github.com/alecheckert/quot; *Heckert, 2022*) with default settings. Custom MATLAB code (https://gitlab.com/tgwgraham/papacode_v1; *Graham, 2022c*) was used to extract all trajectory segments occurring within the first 30 frames after pulses of 405 nm light (DR trajectories) and 561 nm light (PAPA trajectories). PAPA and DR trajectories were then separately analyzed using a Bayesian 'fixed-state sampler' algorithm (https://github.com/alecheckert/spagl; *Heckert et al., 2022*), which estimates the posterior probability distribution over a fixed array of diffusion coefficients. To allow a side-by-side comparison of the distributions for PAPA and DR trajectories, the same number of trajectories was included in the analysis

for each. To this end, trajectories were randomly subsampled without replacement from whichever condition, PAPA or DR, had more trajectories. The fraction bound was calculated in *Figure 4c and f* and *Figure 4—figure supplement 3c and f* by reanalyzing the data using a reduced two-state model with diffusion coefficients 0.01 and 8.3 $\mu m^2$/s. The fraction slow-diffusing was calculated in *Figure 4i and l* and *Figure 4—figure supplement 3i* by fitting to a three-state model with diffusion coefficients 0.01, 2.1, and 13.2 $\mu m^2$/s (*Figure 4i*) or 0.01, 1.3, and 15.8 $\mu m^2$/s (*Figure 4l*, *Figure 4—figure supplement 3i*). Fraction bound for androgen receptor (*Figure 5—figure supplement 1c*) was calculated by fitting to a two-state model with diffusion coefficients 0.01 and 4.4 $\mu m^2$/s. Diffusion coefficients used in the reduced models correspond to the local maxima of the ensemble distributions (*Figures 4b, e, h, k and 5c and d*). Displacement histograms in *Figure 4—figure supplement 2b,d,f,and h* were tabulated using custom code in MATLAB.

A more streamlined and user-friendly Python module for PAPA-SPT analysis, employing an updated version of the SASPT analysis software, will be maintained at https://gitlab.com/tgwgraham/papacode_v2 (copy archived at swh:1:rev:77dbcca3d4ca2833a5d051d495d52444ba34ac27; *Graham, 2022a*).

## PyRosetta simulations

Inter-fluorophore distances were computed from simulated structural ensembles of each linker construct generated using PyRosetta. Crystal structures of Halo (PDB: 6u32), SNAPf (PDB: 6y8p), and titin Ig (PDB: 1tit) were used to model structured regions. Regions lacking density were filled in using RosettaRemodel, and co-crystalized fluorophores bound to Halo and SNAPf were used to estimate inter-fluorophore distance (*Huang et al., 2011*). After filling in missing residues, each structure was minimized using the FastRelax protocol, and starting structures were generated by concatenating structured regions using linkers corresponding to those used in experimental constructs. Ensembles were generated using an adapted version of the FastFloppyTail method used for sampling disordered protein regions, in which only residues comprising the inter-domain linkers were allowed to move (*Ferrie and Petersson, 2020*). The adapted version of the FastFloppyTail algorithm features the addition of the BackrubMover, to allow for motion within a large loop present in SNAPf and facilitate more complete sampling (*Smith and Kortemme, 2008*). After application of the adapted FastFloppyTail protocol, resultant structures were minimized using FastRelax. Each ensemble consisted of 100 structures from which inter-fluorophore distances were computed. Scripts used to generate these ensembles along with the input structures and resultant ensembles can be found at https://github.com/jferrie3/FusionProteinEnsemble, (copy archived at swh:1:rev:9a8ffe946a20d8efb-6c4eb531b22dd96e7431e28; *Ferrie, 2022*).

## Acknowledgements

Thanks to Luke Lavis, Samantha Rider, Alec Heckert, John Lis, Philip Versluis, Joe Loparo, Max Staller, Albert Qin, Viktorija Glembockytė, and the members of the Tjian-Darzacq group for helpful discussions; to Matt Akamatsu for sharing a plasmid encoding the synthetic 60-mer protein; to Vinson Fan for help with androgen receptor cloning; to the UC Berkeley Flow Cytometry Facility for assistance generating clonal cell lines; and to Holly Aaron (UC Berkeley Molecular Imaging Center) and Ana Robles for assistance with microscopy. The microscope used for fluorescence lifetime imaging was purchased with funding from NIH grant S10OD025063. TG was supported by a postdoctoral fellowship from the Jane Coffin Childs Memorial Fund for Medical Research, JF is a Howard Hughes Medical Institute Awardee of the Life Sciences Research Foundation, and RT is an investigator of the Howard Hughes Medical Institute.

## Additional information

### Competing interests

Thomas GW Graham: is an inventor on a pending patent application (PCT/US2021/062616) related to the use of PAPA as a molecular proximity sensor. Xavier Darzacq: is a member of eLife's Board of Directors; is a co-founder of Eikon Therapeutics, Inc; is an inventor on a pending patent application

(PCT/US2021/062616) related to the use of PAPA as a molecular proximity sensor. The other authors declare that no competing interests exist.

### Funding

| Funder | Grant reference number | Author |
| --- | --- | --- |
| Howard Hughes Medical Institute | | Robert Tjian |
| Jane Coffin Childs Memorial Fund for Medical Research | | Thomas GW Graham |
| Life Sciences Research Foundation | | John Joseph Ferrie |

The funders had no role in study design, data collection and interpretation, or the decision to submit the work for publication.

### Author contributions

Thomas GW Graham, Conceptualization, Data curation, Software, Formal analysis, Validation, Investigation, Visualization, Methodology, Writing – original draft, Writing – review and editing; John Joseph Ferrie, Conceptualization, Resources, Software, Investigation, Methodology, Writing – review and editing; Gina M Dailey, Resources; Robert Tjian, Xavier Darzacq, Conceptualization, Supervision, Funding acquisition, Writing – review and editing

### Author ORCIDs

Thomas GW Graham  http://orcid.org/0000-0001-5189-4313
Gina M Dailey  http://orcid.org/0000-0002-8988-963X
Robert Tjian  http://orcid.org/0000-0003-0539-8217
Xavier Darzacq  http://orcid.org/0000-0003-2537-8395

### Decision letter and Author response

Decision letter https://doi.org/10.7554/eLife.76870.sa1
Author response https://doi.org/10.7554/eLife.76870.sa2

---

## Additional files

### Supplementary files

• Transparent reporting form

• Source data 1. Raw data of the plots in *Figure 2*, *Figure 3*, *Figure 4*, and *Figure 5*. fig22_FLIM.mat: MATLAB data file containing calculated fluorescence lifetimes (t1) and Förster resonance energy transfer (FRET) efficiencies (FRETe) for each condition in *Figure 2*. fig22_PAPA.csv: mean and standard error of proximity-assisted photoactivation/direct reactivation (PAPA/DR) ratio as a function of 561 nm pulse duration for each condition in *Figure 2*. fig33_rawdata.mat: single-cell PAPA/DR ratio measurements for all cells in the rapamycin-treated (allrapa) and dimethylsulfoxide (DMSO)-treated control (alldmso) conditions in *Figure 3*. The top row gives the time in minutes before/after treatment, while the lower row gives the PAPA/DR ratio. fig44.zip: subfolders contain data plotted in the corresponding figure panels. Subfolders b, e, h, and k contain two files called 0_rbme_marginal_posterior.csv and 1_rbme_marginal_posterior.csv, which correspond to diffusion spectra of PAPA and DR trajectories, respectively. The first column contains diffusion coefficients, while the second column contains probabilities. Subfolders c, f, i, and l contain the fraction bound (c, f) or fraction slow-diffusing (i, l) for PAPA and DR trajectories from single cells, obtained from analysis with a reduced model. fig5b.mat: single-cell PAPA/DR ratio measurements for all cells in the experiment in *Figure 5b*. The top row gives the time in minutes before/after DHT treatment, while the lower row gives the PAPA/DR ratio. fig5c.mat: diffusion spectra for PAPA and DR trajectories in *Figure 5d*. fig5d.mat: diffusion spectra for PAPA and DR trajectories in *Figure 5d*.

### Data availability

Source data for Figures 2–5 are included in an accompanying zip file.

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

## Appendix 1

### Why not use smFRET?

smFRET provides a powerful way to measure molecular-scale distances in reconstituted mixtures of purified components and in cell-free extracts (*Crawford et al., 2013a*; *Graham et al., 2016*; *Hellenkamp et al., 2018*; *Lerner et al., 2021*). While smFRET has also been used in live cells (*Lerner et al., 2021*; *Sustarsic and Kapanidis, 2015*), these applications have been limited to two categories: (1) measurement of *intra*-molecular conformational changes of molecules double-labeled in vitro and introduced into cells at a very low concentration by electroporation, microinjection, or heat shock (*Crawford et al., 2013b*; *Fessl et al., 2012*; *König et al., 2015*; *Sakon and Weninger, 2010*)—or in one case, expressed at a low level by transient transfection (*Okamoto et al., 2020*). (2) Detection of inter-molecular interactions between membrane proteins expressed at a very low level and imaged by TIRF (*Asher et al., 2021*; *Quast and Margeat, 2021*; *Sako et al., 2000*; *Sotolongo Bellón et al., 2022*; *Wilmes et al., 2020*).

Why is live-cell smFRET restricted to these special cases? The most serious challenge is that single-molecule imaging requires a very low concentration of labeled molecules, far below the endogenous concentration of most proteins (*Hellenkamp et al., 2018*). Sparse labeling provides a way to detect single molecules of one type of protein, but it is much harder to detect protein complexes by simultaneous sparse labeling of two different proteins. For instance, consider two proteins that are each present at a typical concentration of 100,000 molecules per nucleus (*Biggin, 2011*; *Cattoglio et al., 2019*; *Ma et al., 2005*). Unambiguous single-particle tracking of diffusing molecules requires that no more than about 10 molecules per nucleus (order of magnitude) fluoresce at a time—in this example, 1 out of 10,000 molecules. This presents no obstacle to imaging a single species of protein. However, two-color sparse labeling is not a practical way to detect interactions between two proteins because only one out of $(10,000)^2 = 100,000,000$ complexes will be double-labeled. While the exact numbers will of course differ from case to case, sparse double-labeling is not in general an efficient or quantitative way to detect intermolecular interactions within cells.

If instead the sample is sparsely labeled with one fluorophore and densely labeled with the other, spectral crosstalk becomes a serious problem. Because the excitation and emission spectra of fluorophores typically have long tails (*Appendix 1—figure 1a and d*), some donor fluorescence 'bleeds through' into the acceptor channel (*Appendix 1—figure 1a and b*), while the wavelength used to excite the donor also directly excites the acceptor to some extent (*Appendix 1—figure 1d and e*). When the labeled molecules are sparse (e.g., when immobilized on a coverslip in vitro), this crosstalk can be quantified and corrected for (*Hellenkamp et al., 2018*; *Lee et al., 2005*). The crosstalk becomes overwhelming, however, when one of the two channels is densely labeled. Bleed-through from a densely labeled donor (*Appendix 1—figure 1b and c*) or direct-excited fluorescence from a densely labeled acceptor (*Appendix 1—figure 1e and f*) will typically overwhelm the much weaker smFRET signal.

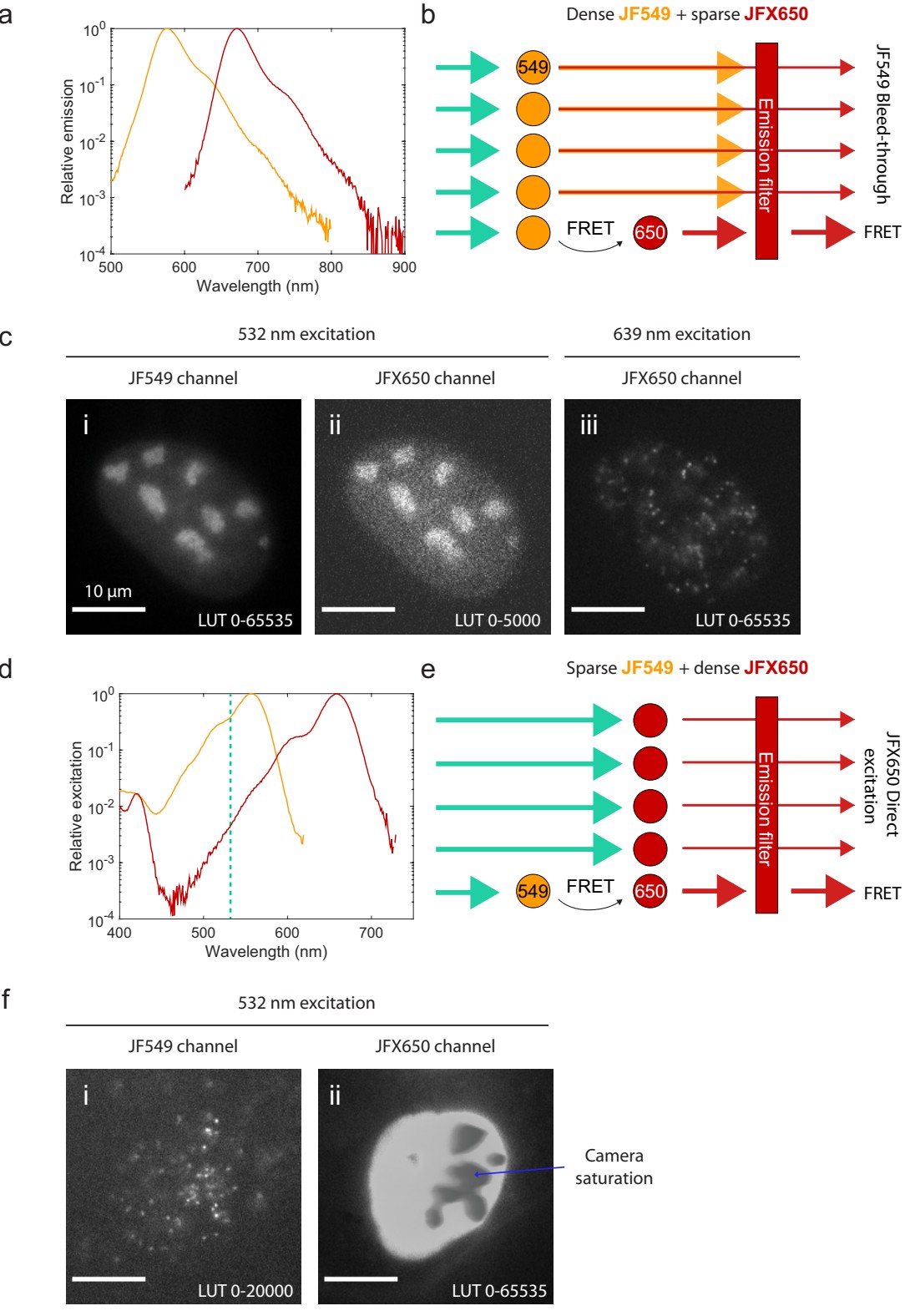

**Appendix 1—figure 1.** Spectral crosstalk impedes the use of single-molecule Förster resonance energy transfer (smFRET) as an interaction sensor in live cells. (**a**) Emission spectra of JF549 (orange) and JFX650 (red). Data from fpbase.org are replotted with a logarithmic y-axis scale to show the long tail of emission. (**b**) Schematic of donor bleed-through. Weak far-red emission from the donor fluorophore (JF549) passes through the acceptor emission filter (JFX650), overwhelming the much weaker FRET signal if the donor is densely labeled and the acceptor is

*Appendix 1—figure 1 continued on next page*

*Appendix 1—figure 1 continued*

sparsely labeled. (**c**) Experimental example of high background due to donor bleed-through. U2OS cells with endogenously tagged NPM1-Halo were double-labeled with a high concentration of JF549 HTL and a lower concentration of JFX650 HTL. Excitation of JF549 with 532 nm light yielded diffuse bleed-through signal in the JFX650 channel (ii), in contrast to the discrete single-molecule spots seen with direct excitation of JFX650 with 639 nm light (iii). (**d**) Absorption spectra of JF549 (orange) and JFX650 (red). Data from fpbase.org are replotted with a logarithmic y-axis scale to show the long tail of absorption. The wavelength used for donor excitation in this figure, 532 nm, is shown as a vertical dashed line. (**e**) Schematic of acceptor direct excitation. While the acceptor fluorophore (JFX650) is excited by 532 nm light more weakly than the donor fluorophore (JF549), this direct excitation can overwhelm the FRET signal if the acceptor is in large excess of the donor. (**f**) Experimental example of high background due to acceptor direct excitation. U2OS cells with endogenously tagged NPM1-Halo were double-labeled with a high concentration of JFX650 HTL and a lower concentration of JF549 HTL. Excitation of JF549 with 532 nm light yielded sparse single-molecule localizations in the JF549 channel but extremely strong, diffuse signal in the JFX650 channel due to JFX650 direct excitation. This background fluorescence was so intense that it saturated the EMCCD camera, leading to artifactual gray splotches in the image (blue arrow).

In contrast, spectral crosstalk is not a problem for PAPA. The sender is excited only briefly, and reactivation is then detected by direct excitation of the receiver, meaning that fluorescence of reactivated receiver molecules is not contaminated by bleed-through fluorescence from the sender. Molecules may be densely labeled with impunity in the sender channel, permitting efficient detection of their interaction with molecules sparsely labeled with the receiver. As a result, PAPA can detect interactions between molecules at concentrations much too high for either two-color SPT or smFRET.

## Appendix 2

### Sources of background in PAPA-SPT

In an ideal PAPA-SPT experiment, single-particle trajectories observed after a green reactivation pulse would consist entirely of complexes in which sender- and receiver-labeled molecules physically interact. Such an idealized pure population of complexes is unattainable in practice due to contributions from several other types of localizations. We describe these background contributions below and suggest how future experimental and data analysis methods might minimize or correct for them.

### 'Leftover' reactivated fluorophores from previous photoactivation pulses

After each photoactivation pulse, a sufficiently long period of red illumination will cause reactivated molecules to photobleach or return to the dark state. However, if reactivated molecules survive between reactivation pulses, then those reactivated by a preceding DR reactivation pulse will contaminate subsequent PAPA trajectories and vice versa. This can be avoided by monitoring the average number of localizations per frame in each experiment to confirm that the red illumination period between reactivation pulses is long enough to restore the number of localizations per frame to baseline. While we imaged U2OS cells, which are flat enough that fluorophores photobleach rapidly throughout their depth, background due to leftover fluorophores may become more troublesome when imaging thicker specimens.

### Spontaneous reactivation

In addition to stimulated reactivation, dark-state fluorophores undergo spontaneous ('thermal') reactivation at a low basal rate, giving rise to a steady background number of localizations per frame. While this is useful for dSTORM imaging (*Tang et al., 2021*), it results in contamination of PAPA trajectories with noninteracting molecules that have reactivated spontaneously.

Because the number of localizations from spontaneous reactivation will presumably increase in proportion to the total number of dark-state fluorophores, this type of contamination is expected to become more severe the higher the concentration of receiver-labeled molecules. Reducing the fraction of molecules labeled with the receiver and increasing the reactivation pulse duration may help to maximize the ratio of photostimulated to spontaneous reactivation.

In the future, analysis algorithms could perhaps be modified to 'subtract' spontaneous reactivation from the diffusion spectrum by using the observed number of localizations before and after photostimulation pulses to infer the relative contributions of spontaneous and stimulated reactivation.

### Nonspecific PAPA

As described in 'Discussion,' several of our results reveal a low level of reactivation by PAPA independent of direct physical association between sender and receiver-labeled molecules. This is not surprising, given that even molecules with uncorrelated random spatial distributions will come together by chance some fraction of the time. This problem may be exacerbated by the large effective distance range of PAPA (*Figure 2*).

Correcting for nonspecific PAPA quantitatively will be challenging, but we suspect it will be possible in at least some cases. Nonspecific reactivation is expected to scale linearly with the concentration of sender. Consistent with this, high concentrations of free JF549 dye induced nonspecific PAPA of JFX650-SNAPf, which increased in proportion to the JF549 concentration (*Appendix 2—figure 1*). In principle, it should be possible to measure the rate of nonspecific PAPA as a function of JF549-Halo concentration and then use this standard curve to estimate the rate of nonspecific PAPA corresponding to the concentration of the sender-labeled protein of interest. Alternatively, paired cell lines could be constructed that express the same concentration of Halo-tagged protein with either a SNAPf-tagged interacting protein or noninteracting SNAPf control protein (in the same subcellular compartment) to measure the rate of background reactivation. The Bayesian analysis framework used here could perhaps be extended to infer the probability that a given trajectory arises from specific or nonspecific PAPA, based on the relative rates of these two processes. Correcting for nonspecific PAPA will likely be simplest for proteins with a uniform spatial distribution, while more sophisticated models may be required to account for background PAPA between proteins with substantial spatial heterogeneity.

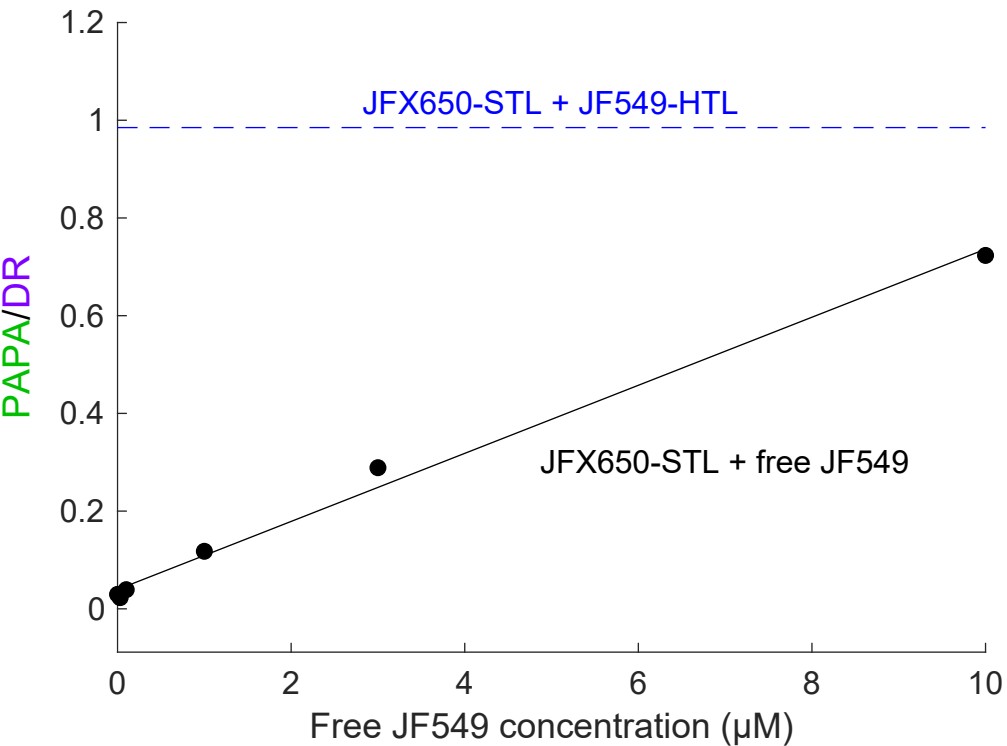

**Appendix 2—figure 1.** Background reactivation by free JF549 dye. U2OS cells expressing Halo-SNAPf-3xNLS were labeled with JFX650-STL only and incubated with various concentrations of free JF549 dye. PAPA/DR ratio (black points) was measured as in the main text figures by calculating the ratio of reactivation due to 7 ms pulses of 561 nm and 405 nm light. Solid black line shows a linear fit. For comparison, the dashed blue line shows the PAPA/DR ratio measured on the same day for Halo-SNAPf-3xNLS double-labeled with JFX650-STL and JF549-HTL.

## Nonspecific labeling by JFX650 SNAP ligand

When analyzing PAPA-SPT experiments, we occasionally noticed a minor peak at D = 0.3–0.4 μm²/s in the diffusion spectra of spontaneously reactivated trajectories, that is, those that did not occur immediately after a green or violet photostimulation pulse (*Appendix 2—figure 2a and b*, orange arrows). The occurrence of a peak with similar D for samples with different SNAPf-tagged proteins suggested that this component was nonspecific. Indeed, when we analyzed weak background JFX650 STL staining of U2OS cells not expressing a SNAPf-tagged protein, we observed a peak with a similar diffusion coefficient, along with a smaller, faster-diffusing peak (D = 3–4 μm²/s; *Appendix 2—figure 2d*, black curve). Thus, in addition to specifically labeling SNAPf-tagged proteins, JFX650-SNAP tag ligand produces at least two types of background staining in U2OS cells with well-defined diffusion coefficients. (As a more optimistic aside, detection of these background components highlights the ability of our state array SPT algorithm to identify subpopulations that might go unnoticed by models with a predefined number of states.)

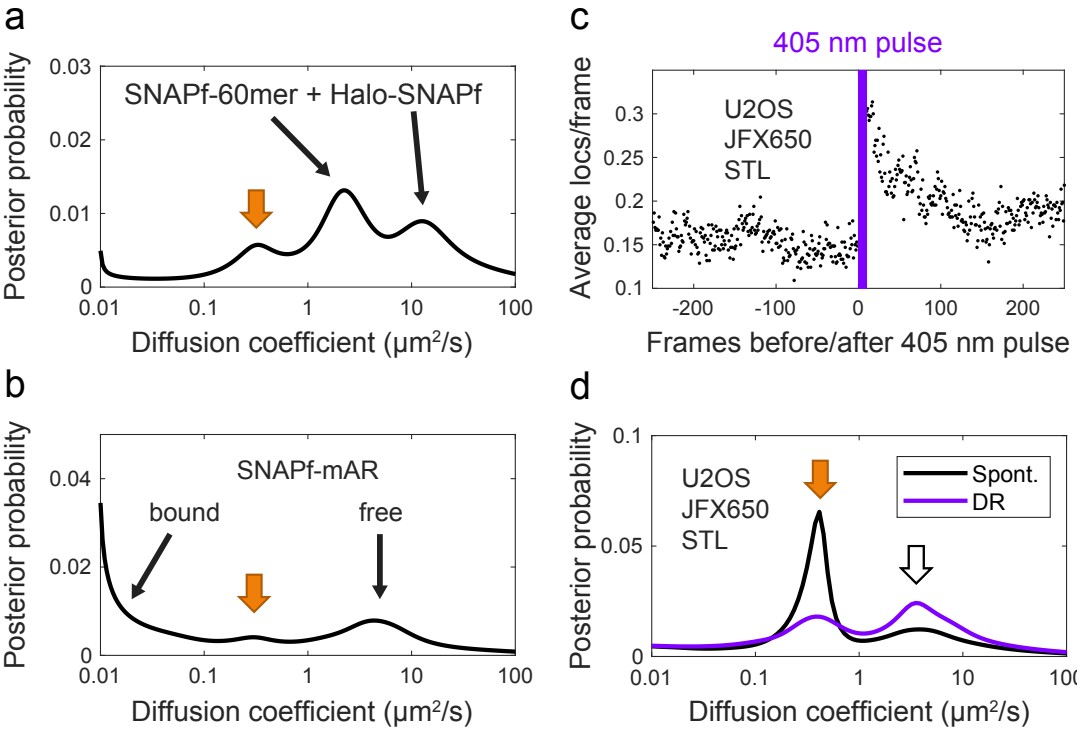

**Appendix 2—figure 2.** SNAP ligand background staining. (**a, b**) Two example diffusion spectra of spontaneously reactivated trajectories in which an additional unidentified peak was observed at ~0.3–0.4 µm²/s (orange arrow). (**a**) SNAPf-60mer + Halo-SNAPf 2-component mixture. (**b**) SNAPf-mAR + Halo-mAR with 10 nM dihydrotestosterone (DHT). (**c**) 405 nm reactivation of JFX650 STL background staining in U2OS cells not expressing a SNAPf-tagged protein, averaged over 10 cycles of photostimulation. (**d**) Diffusion spectrum of JFX650 STL background staining in U2OS cells not expressing a SNAPf-tagged protein. Background staining consists of distinct slow-diffusing (orange arrow, ~0.3–0.4 µm²/s) and fast-diffusing (white arrow, ~3–4 µm²/s) components.

Reactivation of JFX650 STL background staining was observed in response to 405 nm light (*Appendix 2—figure 2c*), and 405-nm-reactivated trajectories were enriched for the fast-diffusing peak compared to the slow-diffusing peak (*Appendix 2—figure 2d*), suggesting a difference in the chemical environment of the fluorophore in the two background species. One endogenous protein known to be labeled by SNAP ligand—albeit at <1% the rate of SNAPf—is $O^6$-methylguanine DNA-methyltransferase (MGMT), the human enzyme from which SNAP tag was originally derived (*Mollwitz et al., 2012*). SDS-PAGE of lysates from JF549 STL-stained U2OS cells revealed a background band consistent in size with MGMT (~22 kDa), along with a second, lower-molecular weight band (*Figure 2—figure supplement 1*), although we have not yet determined the identities of these nonspecifically labeled proteins. Nonspecific dye association with membranes is another potential source of background staining (*Hughes et al., 2014*).

Because the SNAPf-tagged proteins we imaged in this article were expressed at fairly high levels, specific labeling was much greater than nonspecific staining, and nonspecific peaks were only rarely observed in diffusion spectra. However, this form of background staining could pose a more serious problem for SNAPf-tagged proteins expressed at a lower level. Development of improved combinations of self-labeling tags, such as recently described orthogonal HaloTag variants (*Kompa et al., 2022*), will be important for the continued technical advancement of both PAPA and SPT in general.

## Appendix 3

### Supplementary note 1

Comparison of the SDS-PAGE gels in *Figure 2—figure supplement 1c and d* shows that the Halo component of the self-cleaving Halo-PT2A-SNAPf fusion is present at a higher concentration than the SNAPf component. This is expected, given that ribosomes often terminate translation at self-cleaving peptide sequences without restarting translation of the next open reading frame (1). Although we attempted to flow-sort populations of cells expressing similar levels of Halo, the Halo component of Halo-PT2A-SNAPf is expressed at a somewhat higher level than the other constructs (*Figure 2—figure supplement 1c*). Thus, the background PAPA rate observed for Halo-PT2A-SNAPf may overestimate the contribution of nonspecific background to the PAPA rate of the other constructs.

SDS-PAGE analysis of Halo-Ig-SNAPf linker constructs stained with Halo ligand revealed both a full-length protein and a regular ladder of faster-migrating bands, whose molecular weights correspond to Halo fused to different numbers of Ig repeats (*Figure 2—figure supplement 1c*). SNAP ligand, in contrast, predominantly labeled the full-length protein (*Figure 2—figure supplement 1d*); smaller fragments were only faintly visible with enhanced image contrast (*Figure 2—figure supplement 1d*, lower panel). Because the FRET donor (JF549) and PAPA receiver (JFX650) were conjugated to SNAPf, which is almost exclusive to full-length polypeptides, we expect that the presence of smaller Halo-tagged fragments will not impact our measurements of FRET and PAPA efficiency, as these Halo-only fragments will be "invisible" in measurements of JF549-STL fluorescence lifetime and JFX650-STL reactivation.

### Supplementary note 2

In principle, accurate quantification of FRET or PAPA does not require that the FRET donor or PAPA receiver be completely labeled. However, it is critical to thoroughly label the FRET acceptor or PAPA sender, as under-labeling would produce molecules labeled with donor or receiver only (i.e., with no FRET or PAPA), leading to an underestimate of the FRET or PAPA efficiency. Because Halo labeling is much more efficient than SNAPf labeling, we therefore labeled Halo with the acceptor (JFX650-HTL) in FRET experiments and the sender (JF549-HTL) in PAPA experiments, while labeling SNAPf with the opposite fluorophore.

