## [Editor Report]

This work develops a new method to probe protein–protein interactions using proximity-assisted photo activation, in which a receiver fluorophore (longer wavelength) can be photoactivated by the excitation of a nearby sender fluorophore (shorter wavelength). This new method is validated through in-depth characterization, comparison with FRET, and application to known systems of protein–protein interactions. It will expand the tool kit for probing protein–protein interactions.

---

## [Decision Letter]

**Decision letter after peer review:**

Thank you for submitting your article "Proximity-assisted photoactivation (PAPA): Detecting molecular interactions in live-cell single-molecule imaging" for consideration by *eLife*. Your article has been reviewed by 3 peer reviewers, and the evaluation has been overseen by a Reviewing Editor and Anna Akhmanova as the Senior Editor. The following individuals involved in the review of your submission have agreed to reveal their identity: J. Christof M. Gebhardt (Reviewer #2); Helge Ewers (Reviewer #3).

Essential revisions:

1) Please provide a side-by-side comparison of PAPA with smFRET for at least one set of experiments conducted in Figures 3, 4, or 5. For example, will the population percentages or diffusion coefficients (bound vs. unbound) detected using the two methods be the same or different? And why?

2) Please address the issue of how one could distinguish PAPA from DR at the level of single-molecule trajectories instead of relying on the statistics of population measurements or PAPA/RA ratios. This characterization could be done by using in vitro single-molecule imaging where the probability of PAPA and DR on the same molecule could be quantified.

3) Please provide a comparison either experimentally or textually of how PAPA detects protein-protein interactions in comparison with BiFC and other commonly used protein FRET sensors.

*Reviewer #1 (Recommendations for the authors):*

As the new method, PAPA basically benchmarks smFRET in the detection of molecular interactions in live cells, my comments and questions are mainly related to the advantage of PAPA over smFRET.

1. In the introduction, the authors comment that smFRET has proven technically challenging in cells due to the requirement for sparse double-labeling, the large size of genetically encoded tags relative to the working distance of FRET, and the brief observation time (tens of milliseconds) for fast-diffusing complexes. Regarding the requirement for sparse double-labeling, the authors propose that in PAPA, one interacting partner can be sparsely labeled with the receiver and the other densely labeled with the sender, permitting efficient detection of double-labeled complexes. While such labeling strategy can circumvent the tradeoff between labeling density and spectral crosstalk inherent in smFRET, it would increase the unspecific photoactivation in PAPA.

2. PAPA can operate at a longer average intermolecular distance than FRET (Figure 2). While such property may be used to decrease the potential interference from the fusion tag by elongating the linkers between Halo/SNAPf and the protein of interest, it could increase the unspecific photoactivation in PAPA.

3. Regarding "the brief observation time (tens of milliseconds) for fast-diffusing complexes" for smFRET, I think the authors need more characterization to demonstrate the advantage of PAPA in single-molecule measurements. Exploring the capacity of PAPA in detecting single molecules would make this work much more valuable: "Most proteins function by interacting with other proteins, yet we lack tools to study these potentially transient interactions at single-molecule resolution in live cells." Characterization of single-molecule detection may be conducted in vitro. In proof-of-concept experiments, PAPA detected the expected correlation between androgen receptor self-association and chromatin binding at the single-cell level. Single-molecule application could be conducted on nuclear pore complexes.

4. Overall, it is important to perform a direct comparison between PAPA and smFRET in protein-protein interaction measurements, including SMT, sub-population, and interaction dynamics. Actually, as PAPA is based on JF549 and JFX650, this pair of dyes not only be used as "sender" fluorophore and "receiver" fluorophore in PAPA but also be can be used as donor and receptor in smFRET at the same time. Therefore, besides detecting the ratio of green to violet reactivation, FRET signal could be also measured for experiments in Figures 3, 4, and 5, and such comparison is important for reinforcing the advantage of PAPA over smFRET in detecting dynamic protein-protein interactions in live cells, also essential for comparing unspecific activation in PAPA and crosstalk in smFRET.

5. Regarding the physical mechanism underlying PAPA, the authors propose a hypothesis that the excited sender reacts with some other molecule in the cell, producing a short-lived chemical species that diffuses a limited distance to react with and reactivate the receiver dark state. In Supplementary Figure 9, the authors showed that when JFX650-labeled cells were bathed in high concentrations of free JF549 dye, reactivation by green light occurred in proportion to the JF549 concentration. The activation dependence on the concentration of free senders might support the free radical hypotheses.

6. Lastly, as smBiFC is also an important approach for detecting single-molecule protein-protein interaction in live cells, it is necessary to include a few words in the introduction.

*Reviewer #2 (Recommendations for the authors):*

This is a valuable new method to assess protein-protein interactions in live cells. The paper is easy to read and experiments are performed and described comprehensively.

Methods based on split labels are able to provide comparable single-molecule insight into protein-protein interactions, for example, published in Makhija et al., ACS Chem. Biol, 2021 (https://doi.org/10.1021/acschembio.0c00925) and in particular Shao et al., Communications Biology, 2021 (https://doi.org/10.1038/s42003-021-01896-7).

Please compare PAPA to these methods and detail advantages and disadvantages of each method.

*Reviewer #3 (Recommendations for the authors):*

One straightforward experiment that would greatly improve this manuscript: Create and label transmembrane protein-Halo vs transmembrane protein Halo-SNAPf and then I would like to see dual-color video of the single molecules moving in both channels after DR and PAPA, respectively. And a direct quantification of how many molecules are detected in DR ad PAPA. It is a little strange to see a single molecule fluorescence manuscript without any hint of what the data look like.

---

## [Author Response]

Essential revisions:1) Please provide a side-by-side comparison of PAPA with smFRET for at least one set of experiments conducted in Figures 3, 4, or 5. For example, will the population percentages or diffusion coefficients (bound v.s. unbound) detected using the two methods be the same or different? And why?

The reviewers' comments made us realize that we failed to articulate a critical point: It is not possible to do experiments such as those in Figure 3-5 using smFRET. This was a major motivation for developing an alternative approach.

Previous work has applied smFRET to study (1) *intra-*molecular conformational changes of molecules introduced into cells at a low concentration, or (2) *inter-*molecular interactions between membrane proteins expressed at a very low level and imaged by TIRF. Live-cell smFRET has been restricted to these special cases by the requirement that labeled proteins be at an extremely low (~100 pM) concentration. To our knowledge, no one has ever used smFRET to study inter-molecular interactions between endogenously expressed cytoplasmic or nuclear proteins in live cells, which is one reason that we explored PAPA as an alternative. Providing the requested side-by-side comparison would require an unprecedented experimental breakthrough to overcome the concentration limit of smFRET.

An essential point of our paper, which we now state more explicitly, is that PAPA provides a way of detecting protein-protein interactions in typical cases where smFRET does not work. In Appendix 1, we now discuss in more detail the limitations of smFRET and provide a new figure to illustrate these limitations.

2) Please address the issue of how one could distinguish PAPA from DR at the level of single-molecule trajectories instead of relying on the statistics of population measurements or PAPA/RA rations. This characterization could be done by using in vitro single-molecule imaging where the probability of PAPA and DR on the same molecule could be quantified.

To address this point and Reviewer 3’s comments, we now include a time course experiment with immobilized H2B-Halo-SNAPf molecules labeled with either JFX650 STL alone or both JFX650 STL and JF549 HTL, illustrating JF549-dependent reactivation of single JFX650 fluorophores by 561 nm light and JF549-independent reactivation by 405 nm light (Figure 1—figure supplement 3).

It is important to note that because single-particle tracking data are inherently stochastic, obtaining meaningful information always requires the analysis of statistical ensembles. This is true for any SPT experiment. It is never possible to assign with 100% certainty which diffusive state a short trajectory arose from, or to determine the dissociation rate constant of a single bound molecule, because each observed data point is a random draw from a statistical distribution.

Likewise, in the case of PAPA-SPT, “background” sources of reactivation make it impossible to say with 100% certainty whether an individual trajectory represents a complex in which sender and receiver fluorophores directly interact. We now state this caveat more clearly in the Abstract and Introduction, and we estimate the fold enrichment provided by PAPA in experiments with defined 2-component mixtures (Figure 4—figure supplement 4). In Appendix 2, we now discuss in more detail potential sources of background localizations and suggest future directions for addressing the problem of background.

Though PAPA is not perfect, our results show decisively that it can detect molecular interactions and enrich for protein complexes in which two dyes are in proximity. This discovery opens the door to a fundamentally new biophysical approach for probing molecular interactions in live cells.

3) Please provide a comparison either experimentally or textually of how PAPA detects protein-protein interactions in comparison with BiFC and other commonly used protein FRET sensors.

We now include a textual comparison in the Introduction, Discussion, and Appendix 1.

Reviewer #1 (Recommendations for the authors):As the new method, PAPA basically benchmarks smFRET in the detection of molecular interactions in live cells, my comments and questions are mainly related to the advantage of PAPA over smFRET.

As discussed in Appendix 1, smFRET can only be used to detect molecular interactions in live cells if the interacting partners are present at extremely low concentrations and if they can be labeled to bring donor and acceptor dyes very close together. In this rare, fortunate case, smFRET may be preferable to PAPA, as it is exquisitely specific and reveals the precise distance between the two fluorophores. In most cases, however, live-cell smFRET is simply impossible.

1. In the introduction, the authors comment that smFRET has proven technically challenging in cells due to the requirement for sparse double-labeling, the large size of genetically encoded tags relative to the working distance of FRET, and the brief observation time (tens of milliseconds) for fast-diffusing complexes. Regarding the requirement for sparse double-labeling, the authors propose that in PAPA, one interacting partner can be sparsely labeled with the receiver and the other densely labeled with the sender, permitting efficient detection of double-labeled complexes. While such labeling strategy can circumvent the tradeoff between labeling density and spectral crosstalk inherent in smFRET., it would increase the unspecific photoactivation in PAPA.

This is true. However, as we have shown in multiple experiments (e.g., Figure 2 and 3), specific reactivation is markedly greater than nonspecific reactivation, making it possible to use PAPA to enrich for complexes in which JF549- and JFX650-labeled molecules physically associate. By contrast, the problems with sparse labeling and spectral crosstalk alluded to by the reviewer make smFRET useless in most cases for detecting intermolecular interactions in live cells (see Appendix 1).

2. PAPA can operate at a longer average intermolecular distance than FRET (Figure 2). While such property may be used to decrease the potential interference from the fusion tag by elongating the linkers between Halo/SNAPf and the protein of interest, it could increase the unspecific photoactivation in PAPA.

We agree that the broad effective distance range of PAPA is likely a double-edged sword, as it may contribute to nonspecific reactivation. We now mention this point in Appendix 2.

It is important to note, however, that intensity-based FRET measurements suffer from a different type of background in the form of spectral crosstalk. This background is completely debilitating for smFRET at typical endogenous protein concentrations (see Appendix 1–Figure 1).

3. Regarding "the brief observation time (tens of milliseconds) for fast-diffusing complexes" for smFRET, I think the authors need more characterization to demonstrate the advantage of PAPA in single-molecule measurements. Exploring the capacity of PAPA in detecting single molecules would make this work much more valuable: "Most proteins function by interacting with other proteins, yet we lack tools to study these potentially transient interactions at single-molecule resolution in live cells." Characterization of single-molecule detection may be conducted in vitro. In proof-of-concept experiments, PAPA detected the expected correlation between androgen receptor self-association and chromatin binding at the single-cell level. Single-molecule application could be conducted on nuclear pore complexes.

Please note that all the experiments in Figure 4 and 5 involved tracking single molecules. Montages of single-molecule trajectories are displayed in the middle column of Figure 4—figure supplement 2.

We agree that nuclear pore complexes would provide an interesting context for applications of PAPA. As a simpler experiment, we monitored DR or PAPA on individual H2B-Halo-SNAPf molecules labeled with JFX650 STL with or without JF549 HTL. In Figure 1—figure supplement 3, we provide montages and kymographs in which single immobilized fluorophores enter the dark state and get reactivated, and we report the statistics of reactivation. In addition, we now include sample videos from our other experiments, in which single fluorophores can be seen undergoing shelving and reactivation.

4. Overall, it is important to perform a direct comparison between PAPA and smFRET in protein-protein interaction measurements, including SMT, sub-population, and interaction dynamics. Actually, as PAPA is based on JF549 and JFX650, this pair of dyes not only be used as "sender" fluorophore and "receiver" fluorophore in PAPA but also be can be used as donor and receptor in smFRET at the same time. Therefore, besides detecting the ratio of green to violet reactivation, FRET signal could be also measured for experiments in Figures 3, 4, and 5, and such comparison is important for reinforcing the advantage of PAPA over smFRET in detecting dynamic protein-protein interactions in live cells, also essential for comparing unspecific activation in PAPA and crosstalk in smFRET.

Unlike PAPA, smFRET does not work for proteins expressed at typical intracellular concentrations such as those employed in Figure 3, 4, and 5. While we tried not to dwell on the limitations of smFRET in our original manuscript, we now realize that we did not make this point clearly enough. In Appendix 1 and the associated figure, we explain the technical limitations of live-cell smFRET. We summarize these points below.

Unless interacting molecules are both present at a very low concentration, smFRET presents a catch-22 between efficiency of double-labeling and signal to background. Detecting single molecules (either by widefield imaging or within a confocal detection volume) requires that they be labeled sparsely enough that each diffraction limited spot is occupied by much less than one molecule on average. This isn’t a problem when imaging a single type of protein, because the fraction of molecules labeled can be made as small as necessary. The problem arises in trying to detect interactions between two different proteins. If both proteins are labeled sparsely, then double-labeled complexes will constitute a vanishingly small minority of the total. If the labeled fraction of each protein is *f*, then the fraction of complexes that are double-labeled will be *f*^2^.

It might seem that one could get around this limitation by sparsely labeling one protein with donor or acceptor and densely labeling with the other with the opposite fluorophore. In practice, this approach is stymied by the problem of spectral crosstalk between donor and acceptor. If the donor is densely labeled, then the long tail of its emission spectrum “bleeds through” into the acceptor channel (Appendix 1–Figure 1a-c), overwhelming the much weaker FRET signal. If the acceptor is densely labeled, then direct excitation by the laser used to excite the donor again overwhelms the FRET signal (Appendix 1–Figure 1d-f). These contributions of donor bleed-through and acceptor direct excitation are routinely accounted for in smFRET experiments involving sparse immobilized molecules. However, this is not possible for a sample in which one fluorophore is labeled orders of magnitude more densely than the other.

There are special cases in which single-molecule FRET has been used in live cells, involving either (1) double-labeled biomolecules that were microinjected or otherwise introduced into cells at a very low concentration to look at *intra-*molecular conformational changes or (2) membrane proteins expressed at such a low level that single pairs of molecules were resolvable by TIRF microscopy without the need for sparse labeling. To our knowledge, however, smFRET has never been used to study the interaction between endogenous cytosolic or nuclear proteins in live cells.

5. Regarding the physical mechanism underlying PAPA, the authors propose a hypothesis that the excited sender reacts with some other molecule in the cell, producing a short-lived chemical species that diffuses a limited distance to react with and reactivate the receiver dark state. In Supplementary Figure 9, the authors showed that when JFX650-labeled cells were bathed in high concentrations of free JF549 dye, reactivation by green light occurred in proportion to the JF549 concentration. The activation dependence on the concentration of free senders might support the free radical hypotheses.

We modified the text slightly to emphasize that this reaction-diffusion hypothesis is highly speculative. We also mention a second hypothesis, brought to our attention since our initial submission, involving off-peak energy transfer between the sender and the receiver dark state.

The reviewer is totally correct that a reaction-diffusion mechanism would predict a first-order dependence on free sender concentration. Unfortunately, an energy transfer model would make the same prediction. Careful physical chemistry experiments will be required to distinguish between these and possibly other models.

6. Lastly, as smBiFC is also an important approach for detecting single-molecule protein-protein interaction in live cells, it is necessary to include a few words in the introduction.

We agree that smBiFC provides a valuable complementary approach for studying stable protein complexes. We now include a discussion of smBiFC.

Reviewer #2 (Recommendations for the authors):This is a valuable new method to assess protein-protein interactions in live cells. The paper is easy to read and experiments are performed and described comprehensively.Methods based on split labels are able to provide comparable single-molecule insight into protein-protein interactions, for example, published in Makhija et al., ACS Chem. Biol, 2021 (https://doi.org/10.1021/acschembio.0c00925) and in particular Shao et al., Communications Biology, 2021 (https://doi.org/10.1038/s42003-021-01896-7).Please compare PAPA to these methods and detail advantages and disadvantages of each method.

We now include a discussion of BiFC. The major disadvantage of this approach is that assembly of split proteins is effectively irreversible, perturbing the binding equilibrium between partners that may ordinarily bind and dissociate dynamically.

In contrast, PAPA is not expected to perturb dynamic binding equilibria. Consequently, one interesting future direction will be developing a “pulse-chase” variant of PAPA to measure binding and dissociation kinetics of molecular complexes in live cells. We now mention this idea in the Discussion.

Reviewer #3 (Recommendations for the authors):One straightforward experiment that would greatly improve this manuscript: Create and label transmembrane protein-Halo vs transmembrane protein Halo-SNAPf and then I would like to see dual-color video of the single molecules moving in both channels after DR and PAPA, respectively. And a direct quantification of how many molecules are detected in DR and PAPA. It is a little strange to see a single molecule fluorescence manuscript without any hint of what the data look like.

We attempted the proposed experiment by expressing Halo-SNAPf-tagged membrane proteins at a very low concentration in U2OS cells. Unfortunately, because SNAPf is labeled ~2 orders of magnitude less efficiently than Halo (Wilhelm et al., 2021), we found it difficult to obtain a workable density of sparse localizations in both channels. (Note that inefficient SNAPf labeling is not a problem when SNAPf-tagged proteins are present at higher concentrations, the circumstance for which PAPA is the most useful.)

In Figure 1—figure supplement 3, we present a slightly different experiment in which we sparsely labeled H2B-Halo-SNAPf with JFX650 SNAP tag ligand to resolve single immobilized molecules, while we either labeled Halo with JF549 or left it unlabeled as a negative control. Consistent with our other results, DR of immobilized receiver molecules occurred both with and without JF549, while reactivation by 561 nm light occurred much more frequently in the presence of JF549. We display the fraction of receiver dyes reactivated by each light pulse (top panels in a-f), kymographs of the overall population of molecules (bottom panels in a-f), and montages of single molecules getting shelved and reactivated (g).

In response to the reviewer’s critique, we have also included several supplemental videos to show more clearly what our microscopy data look like. Video 1 shows our initial observation of PAPA with NPM1-Halo protein (related to Figure 1). Video 2 shows an increase in PAPA signal upon rapamycin-induced dimerization of JF549-Halo-FRB and JFX650-SNAPf-FKBP (related to Figure 3). Video 3 shows a PAPA-SPT experiment with one of our defined 2-component mixtures (Figure 4j-l), in which the difference between PAPA and DR trajectories can be seen by eye. Also, please note that example single-molecule trajectories can also be found in the middle column of Figure 4—figure supplement 2.